# Fast-SAM3D: 3Dfy Anything in Images but Faster

Weilun Feng [* 1 2]   Mingqiang Wu [* 1 2]   Zhiliang Chen [3]   Chuanguang Yang [✉ 1]   Haotong Qin [4]   Yuqi Li [5]
Xiaokun Liu [1 2]   Guoxin Fan [1 2]   Libo Huang [1]   Yulun Zhang [6]   Michele Magno [4]   Yongjun Xu [1 7]   Zhulin An [✉ 1]

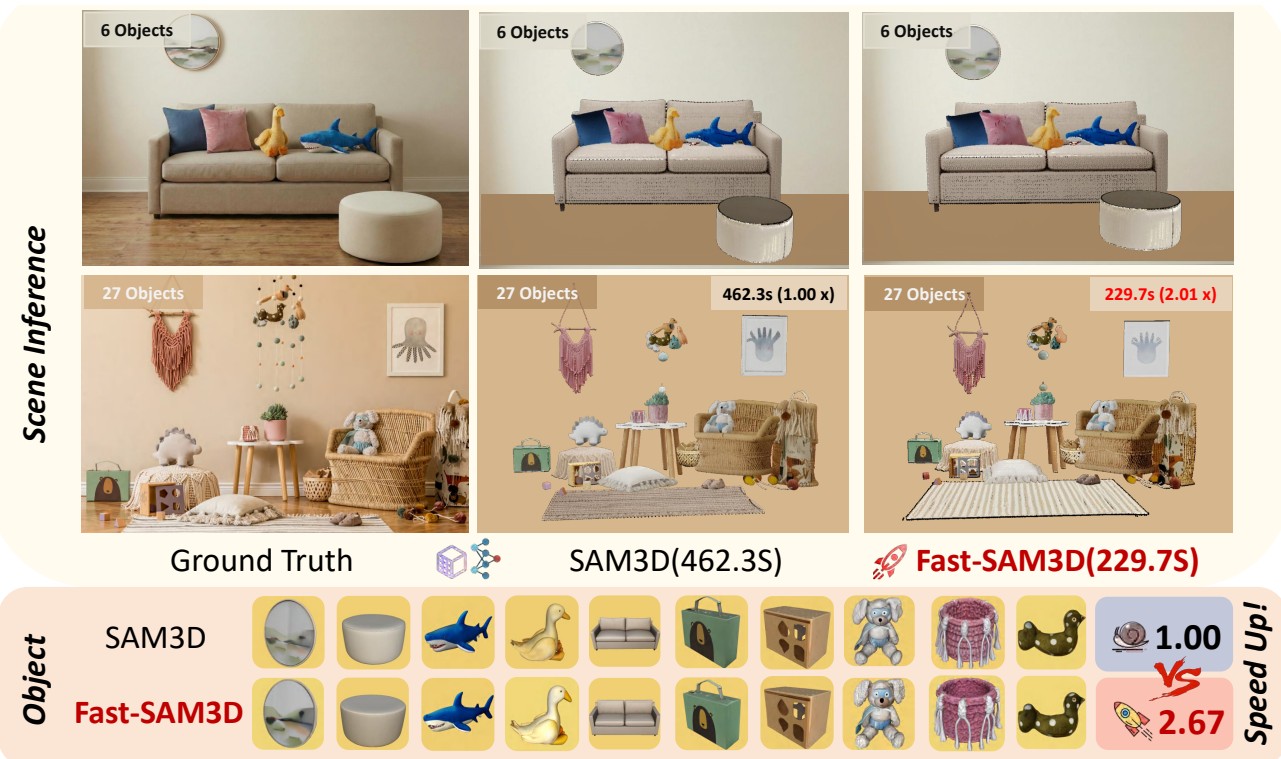

*Figure 1.* **Fast-SAM3D** accelerates the state-of-the-art single-view reconstruction model SAM3D (Chen et al., 2025) by up to **2.67×**, while maintaining the geometric fidelity and semantic consistency.

## Abstract

SAM3D enables scalable, open-world 3D reconstruction from complex scenes, yet its deployment is hindered by prohibitive inference latency. In this work, we conduct the **first systematic investigation** into its inference dynamics, revealing that generic acceleration strategies are brittle

in this context. We demonstrate that these failures stem from neglecting the pipeline's inherent multi-level **heterogeneity**: the kinematic distinctiveness between shape and layout, the intrinsic sparsity of texture refinement, and the spectral variance across geometries. To address this, we present **Fast-SAM3D**, a training-free framework that dynamically aligns computation with instantaneous generation complexity. Our approach integrates three heterogeneity-aware mechanisms: (1) *Modality-Aware Step Caching* to decouple structural evolution from sensitive layout updates; (2) *Joint Spatiotemporal Token Carving* to concentrate refinement on high-entropy regions; and (3) *Spectral-Aware Token Aggregation* to adapt decoding resolution. Extensive experiments demonstrate that Fast-SAM3D delivers up to **2.67×** end-to-end speedup with negligible fidelity loss, establishing a new Pareto frontier for efficient

*Equal contribution  [1]State Key Laboratory of AI Safety, Institute of Computing Technology, Chinese Academy of Sciences  [2]University of Chinese Academy of Sciences  [3]School of Artificial Intelligence, China University of Mining & Technology, Beijing  [4]ETH Zürich  [5]City College of New York, City University of New York, USA  [6]Shanghai Jiao Tong University  [7]Xiamen Institute of Data Intelligence, Xiamen, China. Correspondence to: ✉Zhulin An <anzhulin@ict.ac.cn>, ✉Chuanguang Yang <yangchuanguang@ict.ac.cn>.

*Proceedings of the 43rd International Conference on Machine Learning*, Seoul, South Korea. PMLR 306, 2026. Copyright 2026 by the author(s).

single-view 3D generation. Our code is released in https://github.com/wlfeng0509/Fast-SAM3D.

## 1. Introduction

Unified 3D reconstruction models (Hunyuan3D et al., 2025; Xiang et al., 2025b; Wu et al., 2024; Li et al., 2026; An et al., 2025) that recover high-quality object-centric 3D assets from minimal user input are emerging as a key foundation for scalable 3D perception and content creation. Among them, **SAM3D** (Chen et al., 2025) is distinctive in that it performs *mask-conditioned, open-world* multi-object reconstruction directly from a single scene image, enabling practical reconstruction of arbitrary objects without category-specific training. However, this strong reconstruction quality and generalization comes with substantial computation overhead and severely hinders real-world deployment.

In this work, we conduct the **first systematic investigation** into the inference characteristics of SAM3D. Our profiling reveals that latency is not uniformly distributed but dominated by three coupled components: the dual-stage iterative denoising (structure and texture) and the combinatorial complexity of decoding long token sequences.

Crucially, we find that straightforward applications of generic acceleration techniques like uniform step skipping (Liu et al., 2025; Zhou et al., 2025) or random token pruning (Yang et al., 2025b; Bolya & Hoffman, 2023) are brittle for SAM3D. This arises from the multi-level **heterogeneity** inherent to the pipeline: (i) the *kinematic distinctiveness* between stable shape evolution and sensitive layout updates, where uniform skipping induces pose drift; (ii) the *intrinsic sparsity* of texture refinement, where uniform compute wastes resources on low-entropy surfaces; and (iii) the *spectral variance* across geometries, where instance-agnostic downsampling erases high-frequency details on complex shapes. These observations imply that accelerating SAM3D requires a departure from isolated optimizations toward a model-aware design. To this end, we present **Fast-SAM3D**, a training-free, end-to-end acceleration framework derived from a unified principle: *allocate computation non-uniformly, matching stage-specific difficulty and instance-specific complexity.*

Fast-SAM3D instantiates this principle via three plug-and-play modules that seamlessly integrate into the inference pipeline: (1) **Modality-Aware Step Caching** for the structure generator, which disentangles caching rules to accelerate shape evolution while anchoring sensitive layout attributes; (2) **Joint Spatiotemporal Token Carving** for the latent generator, which eliminates redundancy by concentrating refinement compute solely on dynamically selected active regions; and (3) **Spectral-Aware Token Aggregation** for mesh decoding, which utilizes geometric spectral

entropy to aggressively compress simple shapes while preserving details for complex geometries.

Our contributions are summarized as follows:

- **Systematic Profiling.** We provide the first module-wise characterization of the SAM3D pipeline, identifying key latency sources and revealing why generic acceleration strategies fail due to kinematic and spectral heterogeneity.

- **Holistic Framework.** We propose Fast-SAM3D, a unified, training-free framework that systematically accelerates the geometry, texture, and decoding stages by exploiting their specific redundancies.

- **Adaptive Components.** We design three lightweight modules: modality-aware caching, spatiotemporal token carving, and spectral-aware aggregation. Together, deliver substantial latency reduction while preserving reconstruction quality.

- **Strong empirical results.** Extensive experiments demonstrate significant end-to-end speedups across diverse objects and scenes with negligible degradation in reconstruction fidelity.

## 2. Related Works

**3D Reconstruction and Generation.** Early reconstruction methods focused on deterministically regressing representations like voxels (Xu et al., 2019; Wang et al., 2025c), point clouds (Mildenhall et al., 2021), or meshes (Worchel et al., 2022). While recent feed-forward transformers (Yang et al., 2025a; Wang et al., 2025b; 2024a) achieve rapid inference, but often struggles to high-fidelity generation. The paradigm has notably shifted toward diffusion models (Wang et al., 2025a), where explicit representations (Wu et al., 2024; Xiang et al., 2025b;a) offer superior topological control over implicit counterparts (Li et al., 2024; Yang et al., 2024). However, in the challenging single-view reconstruction, prior works (Lambert et al., 2025; Hunyuan3D et al., 2025; Geng et al., 2025) frequently fail under severe occlusions. **SAM3D** (Chen et al., 2025) overcomes this via mask-conditioned geometric priors for robust open-world reconstruction, yet its prohibitive inference latency remains a significant barrier to interactive deployment.

**Efficient Generative Models.** Acceleration strategies typically fall into training-based methods, such as distillation (Yin et al., 2024; Feng et al., 2025c;a; Dao et al., 2024) and quantization (Li et al., 2023; Feng et al., 2025b;d), or training-free approaches like step optimization (Lu et al., 2025; Ma et al., 2024; Liu et al., 2025) and token pruning (Bolya & Hoffman, 2023; Zou et al., 2024a; Feng

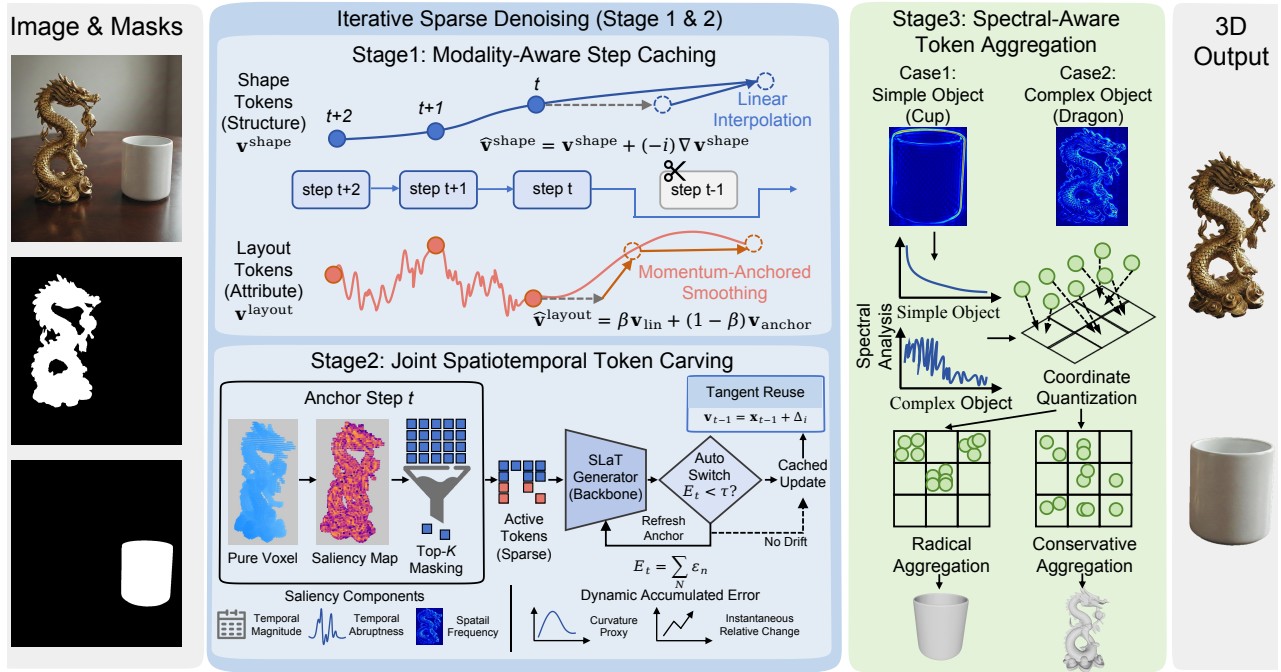

*Figure 2.* **Overview of the proposed Fast-SAM3D framework.** Our approach integrates three heterogeneity-aware modules designed to align computation with the specific dynamics of each stage: (Stage 1) **Modality-Aware Step Caching** disentangles the smooth evolution of shape tokens from the sensitive trajectory of layout tokens; (Stage 2) **Joint Spatiotemporal Token Carving** dynamically eliminates redundancy by concentrating refinement compute solely on high-entropy regions; and (Stage 3) **Spectral-Aware Token Aggregation** adapts the decoding grid density based on the instance-specific geometric complexity.

et al., 2026b). However, these techniques are primarily tailored for 2D domains, exploiting spatial smoothness while neglecting intrinsic *3D structural sparsity* and geometric sensitivity. Existing 3D accelerators are either restricted to implicit fields (Yang et al., 2025c) or regression transformers (Feng et al., 2026a; Shen et al., 2025). Notably, Fast3DCache (Yang et al., 2025b) relies on multi-view redundancy, which is unavailable in single-view tasks. This highlights a critical gap for a *training-free, system-level* framework that accelerates single-view 3D diffusion by simultaneously leveraging temporal consistency and structural sparsity.

## 3. Preliminaries

*Table 1.* Module-wise inference characteristics of SAM3D.

|  | SS Generator | SLaT Generator | Mesh Decoder | Others |
|---|---|---|---|---|
| Param (M) | 1033.63 | 600.43 M | 90.93 M | – |
| Token Length | 5000 | 26892 | 26335 | – |
| Inference Steps | 25 | 25 | – | – |
| Inference Time (ms) | 4090 | 9720 | 13820 | 3370 |
| FLOPs (T) | 95.757 | 219.787 | 324.043 | – |

**SAM3D.** SAM3D (Chen et al., 2025) takes an image $I$ and an object mask $M$ as input, and reconstructs the object's 3D shape $S$, texture $T$, and layout parameters $(R, t, s)$.

**Pipeline and components.** As illustrated in Fig. 3a, SAM3D follows a two-stage coarse-to-fine pipeline: it first predicts coarse structure and global layout, and then refines geometric details and synthesizes texture.

- **Condition embedding** encodes $(I, M)$ into visual tokens using pretrained vision encoders (e.g., DI-NOv2 (Oquab et al., 2023)).

- **Sparse Structure (SS) generator** predicts a coarse structural latent (voxel-like representation) $O$ and global layout $(R, t, s)$ via iterative denoising.

- **Sparse Latent (SLaT) generator** conditions on $(I, M, O)$, and iteratively refines appearance-related signals (e.g., texture/color) and fine-grained geometry.

- **3D decoders** transform refined latents into explicit 3D outputs, including a mesh decoder and a 3D Gaussian splatting (GS) decoder.

**Computation characteristics and inference burden.** Despite strong reconstruction quality and generalization, SAM3D incurs substantial inference overhead. Our profiling results in Tab. 1 and Fig. 3b show that the end-to-end latency is dominated by three components: (i) **SS generator** and (ii) **SLaT generator**, whose costs mainly arise from

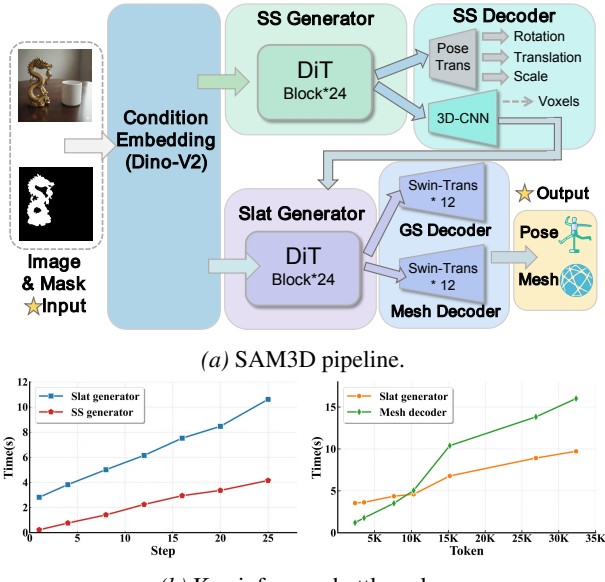

*(a)* SAM3D pipeline.

*(b)* Key inference bottlenecks.

*Figure 3.* **Pipeline characterization and bottleneck analysis. (a)** The standard two-stage coarse-to-fine architecture of SAM3D. **(b)** Latency scaling analysis revealing the dominant computational costs: the linear scaling of iterative denoising steps in the generators and the combinatorial complexity of processing dense voxel tokens in the mesh decoder.

the iterative denoising steps; and (iii) **mesh decoding path**, which becomes expensive when decoding long structured 3D token sequences (e.g., 3D convolution over dense grids). These observations motivate *Fast-SAM3D*: a holistic acceleration framework that jointly reduces the sampling cost in both diffusion stages and the decoding cost in 3D heads, while preserving reconstruction fidelity.

## 4. Methods

### 4.1. Modality-Aware Step Caching for Sparse Structure Generator

**Notation.** We follow the standard diffusion sampling convention (Peebles & Xie, 2023; Liu et al., 2025) where timesteps decrease from $t = T$ (high noise) to $t = 0$ (clean). We denote the current latent by $\mathbf{x}_t$, the diffusion backbone prediction by $\mathbf{v}_t = f_\theta(\mathbf{x}_t, t)$, and a skip of $\Delta$ steps means reusing/predicting the model output at $\mathbf{x}_{t-\Delta}$.

**Problem setting.** In the first stage, the Sparse Structure Generator (SSG) synthesizes coarse 3D shape tokens $\mathbf{v}^{\text{shape}}$ and layout tokens $\mathbf{v}^{\text{layout}}$ (encoding pose/translation/scale) through iterative denoising. We observe pronounced **modality heterogeneity** along the denoising trajectory as shown in Fig. 4: (i) $\mathbf{v}^{\text{shape}}$ evolves relatively smoothly with short-range, near-linear increments; (ii) $\mathbf{v}^{\text{layout}}$ is significantly more volatile, because they directly control the global co-

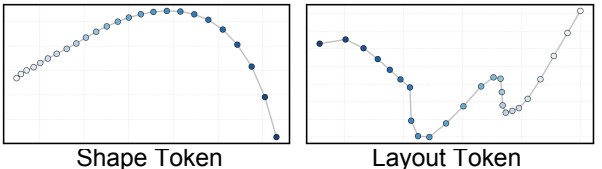

*Figure 4.* **Illustration of modality heterogeneity.** A comparison of update trajectories for shape tokens versus layout tokens. While shape tokens evolve along a smooth path amenable to extrapolation, layout tokens exhibit high-frequency volatility. **More analysis in Appendix Sec. C.**

ordinate frame and thus small errors can induce systematic drift. As a result, applying a uniform caching/prediction policy to all tokens is either unstable (for layout) or overly lagged (for shape).

We therefore propose **Modality-Aware Step Caching**, which disentangles the update rules for $\mathbf{v}^{\text{shape}}$ and $\mathbf{v}^{\text{layout}}$.

**Finite-difference prediction for structural tokens.** Leveraging the smooth evolution of $\mathbf{v}^{\text{shape}}$, we approximate the local trend using a finite difference from two anchor evaluations:

$$\nabla \mathbf{v}_t^{\text{shape}} = \frac{\mathbf{v}_t^{\text{shape}} - \mathbf{v}_{t+k}^{\text{shape}}}{k}, \quad (1)$$

where $k$ is the cache stride. When skipping backbone execution at step $t - i$, we extrapolate structural tokens using 1-order Taylor expansion (Kanwal & Liu, 1989):

$$\hat{\mathbf{v}}_{t-i}^{\text{shape}} = \mathbf{v}_t^{\text{shape}} + (-i) \nabla \mathbf{v}_t^{\text{shape}}. \quad (2)$$

**Momentum-anchored smoothing for layout tokens.** For layout tokens, naive extrapolation is brittle. We introduce an *anchor-corrected* prediction that blends a linear trend with a stable anchor from the most recent full backbone evaluation. Specifically, we first form a linear extrapolation

$$\mathbf{v}_{\text{lin}}^{\text{layout}}(t - i) = \mathbf{v}_t^{\text{layout}} + (-i) \nabla \mathbf{v}_t^{\text{layout}}, \quad (3)$$

and then apply momentum-anchored smoothing:

$$\hat{\mathbf{v}}_{t-i}^{\text{layout}} = \beta \cdot \mathbf{v}_{\text{lin}}^{\text{layout}}(t - i) + (1 - \beta) \cdot \mathbf{v}_{\text{anchor}}^{\text{layout}}, \quad (4)$$

where $\mathbf{v}_{\text{anchor}}^{\text{layout}}$ is the last *full compute* layout token computed by $f_\theta$, and $\beta \in [0, 1)$ is a momentum coefficient. This anchoring suppresses high-frequency jitter and mitigates pose drift, improving temporal consistency of global 3D layout under caching.

### 4.2. Joint Spatiotemporal Token Carving and Adaptive Step Caching

**Motivation: intrinsic refinement sparsity.** In the second stage, the Sparse Latent (SLaT) generator conditions

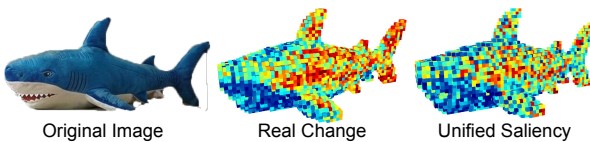

*(a)* Saliency heatmap and corresponding real voxel change.

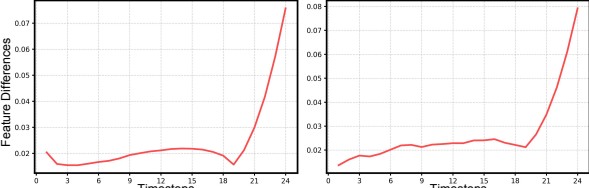

*(b)* Feature difference trajectory visualization.

*Figure 5.* **Visualization of intrinsic refinement sparsity. (a)** Real change map demonstrates that significant updates are spatially sparse, and our unified saliency map accurately predicts this pattern. **(b)** Temporal feature difference plots confirm the diffusion trajectory is non-uniform, validating our dynamic reusing strategy. **More analysis in Appendix Sec. D.**

on the coarse structure $O$ and performs iterative denoising to refine fine-grained geometry and appearance (e.g., color/texture). As shown in Fig. 5a, we observe a systematic efficiency bottleneck, termed **intrinsic refinement sparsity**: token updates are strongly non-uniform across the latent field. Semantically simple regions (low-frequency, smooth surface areas) change slowly across denoising steps, while high-frequency regions (edges, seams, thin structures) exhibit persistent, larger-magnitude updates. Since geometry is largely established by $O$, much of the remaining computation is spent on correcting local high-entropy details, making uniform per-token computation redundant. This motivates adaptive computation in *both* space (which tokens to update) and time (which steps require full evaluation).

### 4.2.1. SPATIOTEMPORAL TOKEN CARVING

**Temporal saliency.** Following the notation used in Sec. 4.1, let $\mathbf{v}_{t,i}$ be the $i$-th output token at step $t$. We quantify per-token temporal activity using first- and second-order variations:

$$\mathcal{M}_i(t) = \|\mathbf{v}_{t,i}\|_2, \qquad \mathcal{A}_i(t) = \|\mathbf{v}_{t,i} - \mathbf{v}_{t+1,i}\|_2, \quad (5)$$

where $\mathcal{M}_i(t)$ measures update magnitude and $\mathcal{A}_i(t)$ highlights abrupt changes.

**Spatial saliency.** To prioritize tokens that contribute to geometric/texture details, we compute a lightweight frequency-based complexity score $\mathcal{S}_{\text{freq}}(i)$ using Fast Fourier Transform (FFT) (Nussbaumer, 1982) statistics, emphasizing tokens with stronger high-frequency structural skeleton.

**Unified importance and carving.** We combine temporal and spatial cues into a unified importance potential:

$$\mathcal{J}_i(t) = \frac{1}{2} \cdot \big(\mathcal{M}_i(t) + \gamma\,\mathcal{A}_i(t)\big) + \frac{1}{2} \cdot \mathcal{S}_{\text{freq}}(i), \quad (6)$$

and keep only the top-$K$ tokens under $\mathcal{J}_i(t)$ as the *active set* at step $t$. This token carving aligns computation with instantaneous detail complexity, substantially reducing redundant updates on low-entropy regions.

### 4.2.2. DYNAMIC ADAPTIVE STEP CACHING

**Curvature-aware trajectory approximation.** As shown in Fig. 5b, beyond spatial sparsity, the diffusion trajectory often contains quasi-linear regimes where the mapping changes smoothly. We estimate local nonlinearity via a curvature proxy (Federer, 1959):

$$\kappa_t = \frac{\|\mathbf{v}_t - \mathbf{v}_{t-1}\|_2}{\|\mathbf{x}_t - \mathbf{x}_{t-1}\|_2}, \quad (7)$$

where $\mathbf{x}_t$ is the input token and smaller $\kappa_t$ indicates a more stable regime in which the trajectory is well approximated by its tangent.

**Tangent update reuse.** Let $i$ denote the most recent *anchor step* where the backbone is fully evaluated. We cache the tangent-like offset

$$\Delta_i := \mathbf{v}_i - \mathbf{x}_i, \quad (8)$$

and, when skipping at step $t$, approximate

$$\hat{\mathbf{v}}_t = \mathbf{x}_t + \Delta_i. \quad (9)$$

**Error-bounded switching.** To prevent uncontrolled drift, we define an instantaneous relative-change proxy

$$\varepsilon_t = \frac{\|\mathbf{v}_t - \mathbf{v}_{t-1}\|_2}{\|\mathbf{v}_{t-1}\|_2} \approx \frac{\kappa_i\,\|\mathbf{x}_t - \mathbf{x}_{t-1}\|_2}{\|\mathbf{v}_{t-1}\|_2}, \quad (10)$$

and accumulate it since the last anchor:

$$E_t = \sum_{n=i+1}^{t} \varepsilon_n. \quad (11)$$

We trigger a full backbone evaluation (refreshing the anchor) when $E_t \geq \mathcal{E}$; otherwise we use the cached tangent update. Formally,

$$\mathbf{v}_t = \begin{cases} f_\theta\big(\mathbf{x}_t \odot \mathbf{m}_t, t\big), & \text{if } E_t \geq \mathcal{E}, \\ \mathbf{x}_t + \Delta_i, & \text{otherwise}, \end{cases} \quad (12)$$

where $\mathbf{m}_t = \mathbb{I}(\mathcal{J}_t \in \text{Top-}K)$ is the binary mask indicating the carved active tokens.

*Table 2.* **Quantitative comparison on SAM3D (Chen et al., 2025) benchmark.** We evaluate reconstruction quality (Visual, Geometric, Layout) and inference speed (Scene Time for all objects in an image and Object Time for a single object) against state-of-the-art acceleration baselines. **Bold**: the best result.

| Method | Visual Alignment | Geometric Accuracy | | | Layout Accuracy | | Acceleration | | | | |
|---|---|---|---|---|---|---|---|---|---|---|---|
| | Uni3D$\uparrow$ | CD$\downarrow$ | $F_1$@0.05$\uparrow$ | vIoU$\uparrow$ | 3D-IoU$\uparrow$ | ICP-rot$\downarrow$ | Scene Time(s)$\downarrow$ | Speed$\uparrow$ | Object Time(s)$\downarrow$ | Speed$\uparrow$ | FLOPs(T)$\downarrow$ |
| SAM-3D | 0.369 | 0.022 | 92.34 | 0.543 | 0.403 | 19.32 | 462.3 | 1.00× | 31.04 | 1.00× | 639.59 |
| *Random Drop* | 0.264 | 0.030 | 83.52 | 0.327 | 0.094 | 32.38 | 402.2 | 1.15× | 15.93 | 1.95 × | 450.42 |
| *Uniform Merge* | 0.329 | 0.023 | 91.48 | 0.540 | 0.367 | 18.44 | 366.8 | 1.26 × | 15.43 | 2.01 × | 251.07 |
| Fast3Dcache | 0.348 | 0.022 | 91.31 | 0.505 | 0.051 | 18.87 | 443.3 | 1.04× | 30.14 | 1.03 × | 532.98 |
| TaylorSeer | 0.344 | 0.028 | 90.95 | 0.504 | 0.374 | 18.43 | 265.6 | 1.74× | 22.93 | 1.35 × | 448.69 |
| EasyCache | 0.342 | 0.028 | 87.06 | 0.432 | 0.186 | 18.44 | 244.9 | 1.89× | 23.11 | 1.34 × | 459.62 |
| **Fast-SAM3D** | **0.350** | **0.022** | **92.59** | **0.552** | **0.375** | **17.71** | **229.7** | **2.01×** | **11.60** | **2.67 ×** | **201.78** |

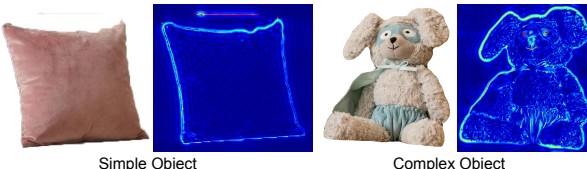

Simple Object          Complex Object

*Figure 6.* **Visualization of instance-level spectral heterogeneity.** Simple objects **(left)** exhibit sparse activations concentrated primarily along boundaries and mostly low frequency, whereas complex objects **(right)** display dense high-frequency components throughout the surface texture. **More analysis in Appendix Sec. E.**

### 4.3. Spectral-Aware Dynamic Token Aggregation for Mesh Decoding

**Motivation.** In the final stage, the mesh decoder transforms the refined sparse latent manifold into mesh primitives, which is runtime-bottlenecked by the *initial* processing over a large number of sparse 3D tokens. While uniform downsampling (Bolya & Hoffman, 2023) is a straightforward remedy, it is instance-agnostic and often destroys fine-grained details for complex shapes. We thus seek an adaptive aggregation policy that conditions token reduction strength on the object's geometric complexity.

#### 4.3.1. SPECTRAL COMPLEXITY ANALYSIS

**Key insight.** We find that a shape's geometric complexity correlates with the distribution of spectral energy as shown in Fig. 6: Simple objects are dominated by low-frequency components, while complex structures exhibit substantial high-frequency energy. This motivates a **spectral-aware** policy that adaptively schedules token aggregation based on an instance-level complexity score.

**Dual-domain spectral metric.** To capture both (i) boundary sharpness from the 2D silhouette and (ii) volumetric topology from the coarse 3D voxel grid, we compute a joint spectral metric from the input mask $M$ and coarse structure $O$. Let $\mathbf{M}_{2D}$ denote the 2D binary mask (from $M$), and let $\mathbf{V}_{3D}$ denote the coarse voxel grid (from $O$). We transform both to the frequency domain:

$$\mathcal{F}_{2D} = \text{FFT2}(\mathbf{M}_{2D}), \qquad \mathcal{F}_{3D} = \text{FFT}(\mathbf{V}_{3D}). \quad (13)$$

**High-frequency energy ratio.** For a signal $\mathbf{X}$, we define the high-frequency energy ratio (HFER) as

$$\mathcal{H}(\mathbf{X}) = \frac{\sum_{\omega \in \Omega_{\text{high}}} \|\mathcal{F}(\mathbf{X})[\omega]\|_2^2}{\sum_{\omega \in \Omega_{\text{total}}} \|\mathcal{F}(\mathbf{X})[\omega]\|_2^2}, \quad (14)$$

where $\Omega_{\text{high}}$ denotes frequencies above a cutoff threshold, and $\Omega_{\text{total}}$ denotes the full spectrum. Larger $\mathcal{H}(\mathbf{X})$ indicates richer high-frequency content and thus higher geometric complexity.

**Composite complexity and adaptive scheduling.** We fuse 2D and 3D complexity into a robust indicator:

$$\mathcal{H}_{\text{joint}} = w \cdot \mathcal{H}(\mathbf{M}_{2D}) + (1 - w) \cdot \mathcal{H}(\mathbf{V}_{3D}), \quad (15)$$

where $w \in [0, 1]$ balances boundary and volumetric cues. We then select an instance-adaptive downsampling factor $\mathcal{S}$ via a discrete schedule:

$$\mathcal{S} = \begin{cases} 1.25, & \mathcal{H}_{\text{joint}} > \tau_{\text{high}}, \\ 1.50, & \tau_{\text{low}} \le \mathcal{H}_{\text{joint}} \le \tau_{\text{high}}, \\ 2.00, & \mathcal{H}_{\text{joint}} < \tau_{\text{low}}, \end{cases} \quad (16)$$

where $\tau_{\text{low}}$ and $\tau_{\text{high}}$ are thresholds.

#### 4.3.2. TOKEN AGGREGATION

**Coordinate quantization.** Given a 3D token centered at $\mathbf{p}_i = (x_i, y_i, z_i)$, we quantize its coordinates with factor $\mathcal{S}$:

$$\hat{\mathbf{p}}_i = \left\lfloor \frac{\mathbf{p}_i}{\mathcal{S}} \right\rfloor. \quad (17)$$

Tokens mapped to the same quantized coordinate are grouped into a voxel-aligned bin, yielding a reduced token set whose size adapts to $\mathcal{S}$.

**Feature aggregation.** For each bin, we aggregate token features using a permutation-invariant operator. To preserve the most prominent features of within each bin, we use max pooling:

$$\hat{\mathbf{z}}(\hat{\mathbf{p}}) = \text{maxpool}_{z_i | i: \ \hat{\mathbf{p}}_i = \hat{\mathbf{p}}} \ \mathbf{z}_i, \tag{18}$$

where $\mathbf{z}_i$ is the latent feature of token $i$ and $\hat{\mathbf{z}}(\hat{\mathbf{p}})$ is the aggregated feature for bin $\hat{\mathbf{p}}$. This reduces the token count by approximately $\mathcal{S}^3$ while preserving salient local features.

## 5. Experiments

### 5.1. Experimental Settings

**Datasets and Metrics.** We adopt SAM3D (Chen et al., 2025) as our base framework. To comprehensively evaluate geometric accuracy and scene layout alignment, we conduct experiments on the **Toys4K** (Stojanov et al., 2021) and **Aria Digital Twin (ADT)** (Pan et al., 2023) datasets. We report standard metrics including Chamfer Distance (CD), F-Score, and Volumetric IoU (vIoU) following protocols in (Liu et al., 2023; Wang et al., 2024b), with meshes aligned via Iterative Closest Point (ICP). Additionally, we assess perceptual fidelity on the **ISO3D** (Ebert, 2025) dataset using the Uni3D score (Zhou et al., 2023) to measure cross-modal semantic consistency.

**Baselines.** We benchmark our method against state-of-the-art diffusion acceleration techniques, categorizing them into: (i) **Token Caching**: Fast3Dcache (Yang et al., 2025b); (ii) **Step Caching**: TaylorSeer (Liu et al., 2025) and Easy-Cache (Zhou et al., 2025). To further validate the efficacy of our proposed saliency-aware mechanisms, we also compare with naive variants, including *Random Drop* for token carving and *Uniform Merge* for mesh decoding.

**See Appendix Sec. A for more detailed descriptions.**

### 5.2. Main Results

Results in Tab. 2 demonstrate that **Fast-SAM3D** achieves the optimal trade-off between efficiency and quality.

**Performance and Efficiency.** Our method delivers significant acceleration, achieving **2.01×** and **2.67×** speedups for scene and object generation, respectively. This substantially outperforms pure step-skipping baselines like TaylorSeer (1.35×) and EasyCache (1.34×), confirming that exploiting spatial redundancy via token carving is essential alongside temporal caching. Remarkably, Fast-SAM3D maintains or even exceeds the base model's geometric fidelity (e.g., F-Score: 92.59 vs. 92.34). We attribute this to the *denoising effect* of our saliency mechanism, which effectively prunes high-frequency noise inherent in the original full-token generation.

**Baseline Analysis.** Comparisons highlight the limitations of existing methods in single-view settings. *Fast3Dcache* brings minimal acceleration (1.03×) due to the lack of multi-view redundancy, while *Random Drop* suffers catastrophic structural collapse (3D-IoU drops to 0.094), proving that 3D structural information is non-uniform and requires our saliency-aware preservation. Fast-SAM3D successfully bridges these gaps, ensuring robust geometry with maximal end-to-end speedup.

**Extended Validation.** Fast-SAM3D is designed as a training-free inference-time acceleration layer rather than a substitute for improving the underlying reconstruction backbone. We further verify this positioning through transfer experiments on TRELLIS (Xiang et al., 2025b) and stage-wise memory profiling in **Appendix Sec. B**, showing that the proposed heterogeneity-aware principle generalizes beyond SAM3D and does not introduce a peak-memory overhead. **Appendix Sec. F** further confirms robustness under 128-view ADT evaluation, degraded masks, and five-run evaluation.

### 5.3. Visual Comparison

Fig. 7 presents a qualitative comparison across diverse categories. **Fast-SAM3D** (2nd Col.) produces results that are perceptually indistinguishable from the original model **SAM3D** (Chen et al., 2025), faithfully preserving both high-frequency textures (e.g., the wooden bird) and global topology. In contrast, generic strategies prove brittle. *Random Drop* (6th Col.) suffers from catastrophic structural collapse, validating that 3D sparsity requires intelligent saliency-aware retention rather than stochastic pruning. Furthermore, step-skipping baselines like TaylorSeer (Liu et al., 2025) exhibit severe *semantic drift*, which completely alters object attributes (e.g., the blue shark turning orange in 5th Col.) due to the unstable evolution of layout tokens. Our method effectively circumvents these pitfalls by anchoring sensitive layout updates, ensuring consistent alignment with the input condition. **We provide more visual comparisons in Appendix Sec. H.**

### 5.4. Ablation Study

We conduct a comprehensive ablation study to validate the individual contributions of our acceleration modules and analyze the sensitivity of key hyperparameters.

**Impact of Acceleration Components.** Tab. 3 disentangles the effects of the three proposed modules: Modality-Aware Step Caching (SS), Joint Spatiotemporal Token Carving (SLaT), and Spectral-Aware Token Aggregation (Mesh). The standalone application of Mesh acceleration yields significant latency reduction (462s → 320s). This identifies the high-resolution mesh decoding process as the computa-

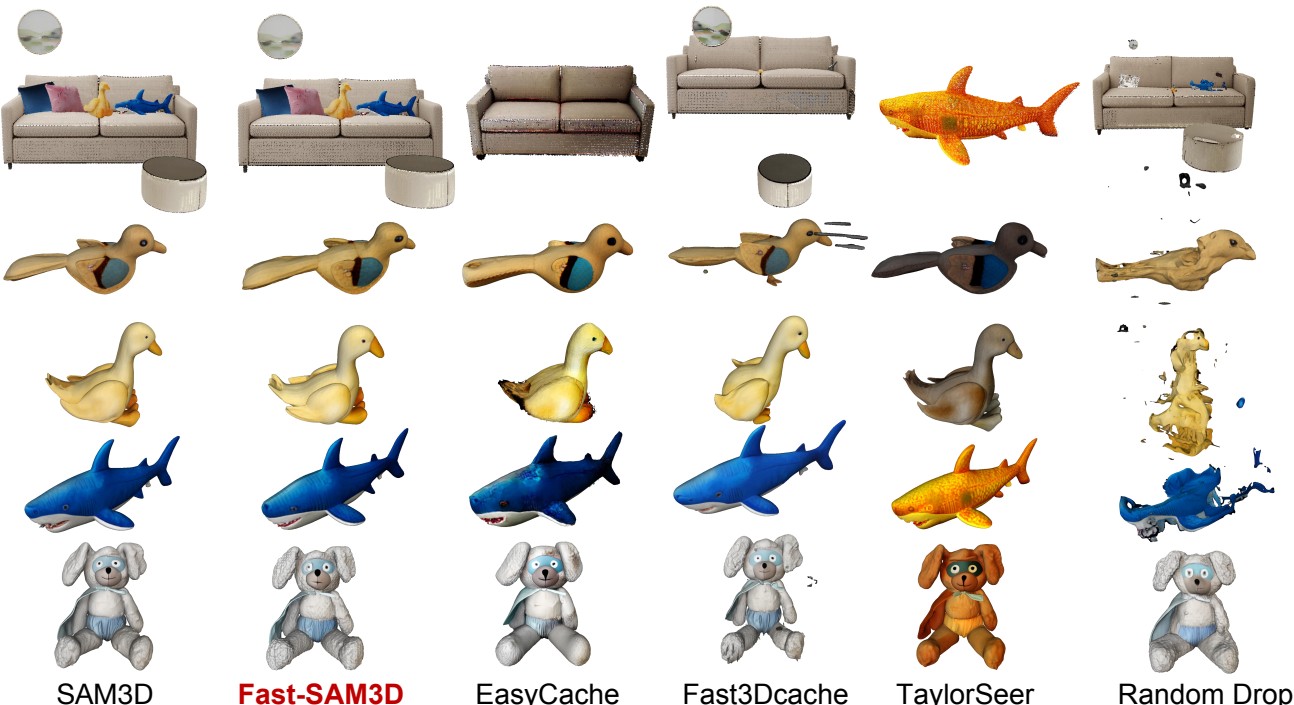

SAM3D     **Fast-SAM3D**     EasyCache     Fast3Dcache     TaylorSeer     Random Drop

*Figure 7.* **Qualitative comparison of our proposed Fast-SAM3D and other methods.**

*Table 3.* Ablation study on different component acceleration.

| Acceleration | | | CD↓ | $F_1$@0.05↑ | vIoU↑ | Scene Time(s)↓ |
|---|---|---|---|---|---|---|
| SS | SLaT | Mesh | | | | |
| ✗ | ✗ | ✗ | 0.022 | 92.34 | 0.543 | 462.33 |
| ✓ | ✗ | ✗ | 0.022 | 92.34 | 0.543 | 408.64 |
| ✗ | ✓ | ✗ | 0.022 | 92.50 | 0.540 | 365.86 |
| ✗ | ✗ | ✓ | 0.022 | 92.43 | **0.557** | 320.41 |
| ✓ | ✓ | ✗ | **0.021** | **92.88** | 0.534 | 310.54 |
| ✓ | ✗ | ✓ | 0.022 | 92.58 | 0.553 | 289.92 |
| ✗ | ✓ | ✓ | 0.022 | 92.43 | 0.554 | 301.34 |
| ✓ | ✓ | ✓ | 0.022 | 92.59 | 0.552 | **229.68** |

*Table 4.* Compact ablations of cache stride $k$ in SS and token-carving percentage $K$ in SLaT. We report key metrics and object inference time.

| Ablation | Setting | $F_1$@0.05↑ | vIoU↑ | 3D-IoU↑ | Object Time(s)↓ |
|---|---|---|---|---|---|
| Cache stride | $k = 2$ | **92.587** | 0.5499 | 0.3750 | 11.85 |
| Cache stride | $k = 3$ | 92.585 | **0.5521** | **0.3750** | 11.60 |
| Cache stride | $k = 4$ | 91.862 | 0.5302 | 0.2408 | 11.39 |
| Cache stride | $k = 5$ | 90.994 | 0.5137 | 0.2331 | 11.17 |
| Carving ratio | top-5% | **92.603** | **0.5589** | 0.3750 | 12.15 |
| Carving ratio | top-10% | 92.585 | 0.5521 | **0.3750** | 11.60 |
| Carving ratio | top-20% | 90.254 | 0.5311 | 0.3640 | **10.91** |

tional bottleneck in the original SAM-3D pipeline, justifying our aggressive spectral-aware aggregation strategy. Interestingly, applying SLaT not only reduces inference time (366s) but also improves geometric fidelity ($F_1$ increases from 92.34 to 92.50). This corroborates our hypothesis that saliency-based carving acts as a spatial filter, removing low-confidence, noisy tokens that would otherwise degrade the final surface. Combining all three modules yields the optimal performance (230s), achieving a $\sim 2.0\times$ speedup without compromising the volumetric intersection over union (vIoU: 0.552 vs. 0.543).

**Cache Stride $k$ (Anchor Update Frequency).** The top block of Tab. 4 studies how frequently SS should refresh the anchor tokens. A short stride ($k = 2$) preserves high fidelity but provides limited acceleration, whereas enlarg-

ing the stride beyond the local linear regime rapidly hurts layout alignment. In particular, $k = 4$ reduces object inference time only slightly but drops 3D-IoU from 0.3750 to 0.2408, indicating accumulated pose drift. We therefore adopt $k = 3$, which achieves the best vIoU (0.5521) and layout accuracy while retaining the desired speedup.

*Table 5.* Ablation study on momentum factor $\beta$ in SS, corresponding to the momentum-anchored smoothing in Eq. 4.

| $\beta$ | CD↓ | $F_1$@0.05↑ | vIoU↑ | 3D-IoU↑ | ICP-rot↓ |
|---|---|---|---|---|---|
| 1.0 | 0.022 | 92.51 | 0.541 | 0.375 | 17.87 |
| 0.9 | 0.022 | 92.58 | 0.551 | 0.375 | 17.97 |
| 0.7 | 0.022 | 92.57 | 0.550 | 0.375 | 17.83 |
| 0.5 | **0.022** | **92.59** | **0.552** | **0.375** | 17.77 |
| 0.3 | 0.022 | 92.58 | 0.551 | 0.375 | **17.71** |

**Momentum Factor $\beta$ (Layout Stability).** Tab. 5 analyzes the layout update coefficient $\beta$. Unlike shape tokens, layout tokens exhibit high volatility. Pure linear extrapolation ($\beta = 1.0$) leads to suboptimal geometric metrics ($F_1$: 92.51) due to cumulative drift. Decreasing $\beta$ increases the influence of the anchor step. We find that $\beta = 0.5$ yields the peak performance in both fine-grained geometry ($F_1$: 92.59) and volumetric completeness (vIoU: 0.552). This suggests that a balanced trade-off that equally weighs the instantaneous linear trend and the stable anchor is crucial for suppressing high-frequency jitter in the global coordinate frame.

**Token-Carving Ratio $K$ (Redundancy in SLaT).** The bottom block of Tab. 4 validates the sparsity assumption behind SLaT. A moderate carving ratio of top-10% preserves the same scene alignment as the conservative top-5% setting while reducing object inference time from 12.15s to 11.60s. However, increasing the ratio to top-20% trades only a marginal additional speed gain for a clear fidelity drop ($F_1$: 92.585 $\rightarrow$ 90.254). This confirms that SLaT should remove redundant updates conservatively: moderate carving exploits spatial sparsity, while overly aggressive carving discards structural details.

*Table 6.* Ablation study on switching threshold $\mathcal{E}$ in SLaT, corresponding to Eq. 12.

| $\mathcal{E}$ | $CD_{\downarrow}$ | $F_1@0.05_{\uparrow}$ | $vIoU_{\uparrow}$ | $3D\text{-}IoU_{\uparrow}$ | $ICP\text{-}rot_{\uparrow}$ | Object Time(s)$_{\downarrow}$ |
|---|---|---|---|---|---|---|
| 1.0 | 0.0216 | 92.575 | 0.5493 | 0.3750 | **17.610** | 11.97 |
| 1.5 | **0.0215** | **92.585** | **0.5521** | **0.3750** | 17.713 | 11.60 |
| 2.0 | 0.0216 | 92.526 | 0.5502 | 0.3750 | 17.982 | **11.46** |

**Switching Threshold $\mathcal{E}$ (Adaptive Refresh).** Tab. 6 studies when SLaT should trigger a full backbone refresh. A conservative threshold ($\mathcal{E} = 1.0$) limits acceleration, while an aggressive one ($\mathcal{E} = 2.0$) introduces mild drift in geometry. We set $\mathcal{E} = 1.5$ as it achieves the best $F_1$ and vIoU while preserving scene alignment, indicating a stable balance between cached updates and anchor refreshes.

*Table 7.* Ablation study of the spectral scheduling thresholds $\tau$ in Spectral-Aware Token Aggregation (Mesh), corresponding to Eq. 16.

| $\{\tau_{\text{low}}, \tau_{\text{high}}\}$ | $CD_{\downarrow}$ | $F_1@0.05_{\uparrow}$ | $vIoU_{\uparrow}$ | Scene Time(s)$_{\downarrow}$ |
|---|---|---|---|---|
| $\{0.0, 0.0\}$ | 0.0208 | 92.884 | 0.5342 | 250.98 |
| $\{0.5, 0.7\}$ | 0.0215 | 92.585 | **0.5521** | 229.68 |
| $\{0.5, 0.9\}$ | 0.0216 | **92.590** | 0.5498 | **226.64** |
| $\{0.3, 0.7\}$ | 0.0214 | 92.527 | 0.5500 | 239.02 |
| $\{0.3, 0.5\}$ | **0.0213** | 92.497 | 0.5485 | 237.26 |

**Merging Thresholds $\tau$ (Speed-Fidelity Trade-off).** Tab. 7 explores the adaptive grid settings $\{\tau_{\text{low}}, \tau_{\text{high}}\}$. While the conservative setting $\{0.0, 0.0\}$ without merging provides a high upper bound for accuracy, it suffers from high latency (250.98s). Our selected configuration $\{0.5, 0.7\}$ identifies the optimal Pareto point: it reduces inference time by $\sim 8.5\%$ while remarkably achieving the highest vIoU (0.5521). This indicates that aggressive downsampling in spectrally simple regions effectively removes redundancy without compromising the object's topological integrity.

**We provide more analysis of modality heterogeneity in Appendix Sec. C, additional studies of saliency prediction and carving factor $\gamma$ in Appendix Sec. D, and spectral complexity weight $w$ in Appendix Sec. E.**

# 6. Conclusion

In this work, we identified and addressed the critical latency bottleneck hindering the interactive deployment of open-world 3D reconstruction frameworks. Through the first systematic analysis of SAM3D's inference dynamics, we revealed that the inefficiency of generic accelerators stems from their inability to accommodate the intrinsic multi-level heterogeneity of 3D generation. In response, we presented **Fast-SAM3D**, a training-free acceleration framework that dynamically harmonizes computational resources with instantaneous generation complexity. By synergizing modality-aware step caching, spatiotemporal token carving, and spectral-aware aggregation, our method successfully decouples structural evolution from redundant computation. Extensive experiments demonstrate that Fast-SAM3D achieves a remarkable **2.67$\times$** speedup while preserving, and in some cases enhancing, geometric fidelity. We hope this work establishes a new baseline for efficient single-view 3D generation and inspires future research into heterogeneity-aware optimization for complex diffusion pipelines.

## Acknowledgements

This work is supported by the National Natural Science Foundation of China under Grant Number 62476264 and 62406312, the Beijing Natural Science Foundation under Grant Number 4244098, the Science Foundation of the Chinese Academy of Sciences, and the Swiss National Science Foundation (SNSF) project 200021E_219943 Neuromorphic Attention Models for Event Data (NAMED).

## Impact Statement

This paper presents work whose goal is to advance the field of machine learning. There are many potential societal consequences of our work, none of which we feel must be specifically highlighted here.

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

# A. Detailed Experimental Settings

## A.1. Data Preparation and Evaluation Protocols

**Geometry Evaluation (Toys4K).** For fine-grained geometry assessment, we utilize the Toys4K dataset (Stojanov et al., 2021). We preprocess the data by rendering ground-truth meshes from randomly sampled viewpoints, applying background removal, and strictly filtering out low-quality samples. This results in a curated test set of 600 unique views (one view per object). We evaluate the reconstruction quality using Chamfer Distance (CD), F-Score ($F_1$@0.05), and Volumetric IoU (vIoU). Note that all generated meshes are rigidly aligned to the ground truth using Iterative Closest Point (ICP) before metric calculation to disentangle pose errors from shape reconstruction quality.

**Scene Layout Alignment (ADT).** To evaluate the model's ability to handle complex scene layouts, we select the Aria Digital Twin (ADT) dataset (Pan et al., 2023). We filter highly similar video frames and select four sequences that represent distinct scene types. From each sequence, we sample 4 key views, yielding 16 evaluation views in total. We report 3D IoU and rotation error (ICP-rot) to measure the alignment accuracy between the reconstructed scene and the ground truth.

**Perceptual Fidelity (ISO3D).** For wild objects lacking 3D ground truth, we employ the ISO3D dataset (Ebert, 2025), consisting of 101 synthetic objects with 101 unique views. Since standard geometric metrics are inapplicable, we use the multi-modal Uni3D (Zhou et al., 2023) model to compute the perceptual similarity score. Specifically, we sample 8,192 points uniformly from the surface of each generated mesh to construct a point cloud. We then calculate the cosine similarity between the point cloud embeddings and the input image embeddings, serving as a proxy for shape-appearance consistency.

## A.2. Details of Evaluation Metrics

We define the specific metrics used to assess geometric fidelity, structural alignment, and perceptual quality. In the following definitions, let $\mathcal{G}$ denote the ground-truth 3D representation and $\mathcal{P}$ denote the generated 3D representation.

**Chamfer Distance (CD, $\downarrow$).** Chamfer Distance measures the bi-directional geometric discrepancy between the generated mesh and the ground truth. We sample $N = 10,000$ points from the surface of both meshes to obtain point clouds $\mathcal{S}_{\mathcal{P}}$ and $\mathcal{S}_{\mathcal{G}}$. The metric is defined as:

$$\mathrm{CD}(\mathcal{S}_{\mathcal{P}}, \mathcal{S}_{\mathcal{G}}) = \frac{1}{|\mathcal{S}_{\mathcal{P}}|} \sum_{x \in \mathcal{S}_{\mathcal{P}}} \min_{y \in \mathcal{S}_{\mathcal{G}}} \|x - y\|_2^2 + \frac{1}{|\mathcal{S}_{\mathcal{G}}|} \sum_{y \in \mathcal{S}_{\mathcal{G}}} \min_{x \in \mathcal{S}_{\mathcal{P}}} \|y - x\|_2^2. \tag{19}$$

Lower CD indicates higher geometric fidelity and closer surface alignment.

**F-Score ($F_1$@$\tau$, $\uparrow$).** While CD is sensitive to outliers, F-Score provides a robust evaluation of reconstruction completeness and precision. It is the harmonic mean of precision and recall at a specific distance threshold $\tau$:

$$F_1(\tau) = \frac{2 \cdot \mathrm{Precision}(\tau) \cdot \mathrm{Recall}(\tau)}{\mathrm{Precision}(\tau) + \mathrm{Recall}(\tau)}, \tag{20}$$

where Precision($\tau$) is the percentage of points in $\mathcal{S}_{\mathcal{P}}$ within distance $\tau$ of any point in $\mathcal{S}_{\mathcal{G}}$, and vice versa for Recall($\tau$). We set $\tau = 0.05$ following (Wang et al., 2024b). Higher F-Score implies better surface coverage and fewer artifacts.

**Volumetric IoU (vIoU, $\uparrow$).** To evaluate the holistic 3D structure and occupancy, we voxelize both the ground truth and generated meshes into a resolution of $32^3$. Volumetric IoU is calculated as the intersection over union of the occupied voxels:

$$\mathrm{vIoU} = \frac{|\mathcal{V}_{\mathcal{P}} \cap \mathcal{V}_{\mathcal{G}}|}{|\mathcal{V}_{\mathcal{P}} \cup \mathcal{V}_{\mathcal{G}}|}, \tag{21}$$

where $\mathcal{V}$ denotes the set of occupied voxels. This metric is particularly useful for assessing whether the model captures the correct global topology and volume.

**ICP Rotation Error (ICP-rot, $\downarrow$).** For the ADT dataset, we focus on the alignment of the object/scene within the global coordinate system. We perform Iterative Closest Point (ICP) alignment and record the magnitude of the rotation component of the transformation matrix required to align the generated shape to the ground truth. A lower ICP-rot error indicates that the model has successfully preserved the correct canonical pose and scene layout without significant drift.

**Uni3D Score (Perceptual Fidelity, ↑).** For the ISO3D dataset where ground-truth 3D models are unavailable, we utilize Uni3D (Zhou et al., 2023), a scalable 3D foundation model aligned with CLIP image embeddings, to measure semantic consistency. We compute the cosine similarity between the embedding of the input image $I$ and the embedding of the generated point cloud $\mathcal{P}$:

$$\text{Score}_{\text{Uni3D}} = \frac{E_I(I) \cdot E_P(\mathcal{P})}{\|E_I(I)\|_2 \|E_P(\mathcal{P})\|_2}, \tag{22}$$

where $E_I$ and $E_P$ are the image and point cloud encoders, respectively. A higher score reflects better preservation of semantic attributes and visual appearance consistent with the input view.

### A.3. Implementation Details

For fair comparison, all baselines are implemented on the same SAM3D backbone and all the experiments are conducted on a single NVIDIA-A800.

- **Fast-SAM3D**: We adopt our proposed Fast-SAM3D by comprehensively accelerating SS Generator, SLaT Generator, and Mesh Decoder. In the SS Generator configuration, we set the cache stride to $k = 3$ and the momentum factor to $\beta = 0.5$, with a warmup period of 2 steps (common setup in diffusion step caching (Zou et al., 2024b; Liu et al., 2025; Zhou et al., 2025)). For the SLaT Generator, the switching threshold is set to $\mathcal{E} = 1.5$ with 2 warmup steps; additionally, the temporal carving factor is configured as $\gamma = 0.7$, and we cache top $0.1\times$ tokens for spatiotemporal carving. Finally, regarding the merging thresholds, we use $\tau_{low} = 0.5$ and $\tau_{high} = 0.7$, with $w = 0.9$.

- **Fast3Dcache** (Yang et al., 2025b): Adapted for single-view inputs by removing its multi-view dependency while retaining its token caching mechanism. As Fast3Dcache targets reducing shape generation tokens, we adopt Fast3Dcache to the SS Generator. We follow the official settings (Yang et al., 2025b) for other parameters.

- **TaylorSeer** (Liu et al., 2025): TaylorSeer reduces diffusion sample steps using feature cache. We adopt TaylorSeer to both SS and SLaT Generator (following the same step caching protocol as ours). In both the SS and SLaT Generator configuration, we also adopt a warmup period of 2 steps for fair comparison.

- **EasyCache** (Zhou et al., 2025): Similar to TaylorSeer, we also adopt EasyCache to both SS and SLaT Generator. Due to the collapsed performance, we use a warmup period of 3 steps for the SS Generator and 2 warmup steps for SLaT Generator. We follow the official settings (Zhou et al., 2025) and maintain a similar acceleration ratio as our step caching setting.

- **Naive Baselines**: *Random Drop* replaces our saliency-based carving (SLaT Generator) with random selection at the same sparsity ratio; *Uniform Merge* replaces our spectral-aware aggregation (Mesh Decoder) with a fixed uniform grid downsampling $S = 2$.

## B. Extended Validation

In this section, we provide additional experiments that further clarify the scope and deployment properties of Fast-SAM3D. These results complement the main paper by evaluating transferability beyond SAM3D and stage-wise memory usage.

### B.1. Transferability to TRELLIS

To examine whether the proposed acceleration principle is tied to SAM3D, we migrate Fast-SAM3D to TRELLIS (Xiang et al., 2025b) and refer to the resulting variant as **Fast-TRELLIS**. Since the first-stage sparse-structure generator in TRELLIS does not explicitly model spatial layout tokens, we replace the layout-aware branch of MASC with TaylorSeer-style step caching in this stage, while transferring the remaining heterogeneity-aware designs to the corresponding latent refinement and decoding stages. As shown in Tab. 8, Fast-TRELLIS reduces TRELLIS inference time from 7.68s to 3.40s, achieving a $2.26\times$ speedup while keeping the main geometry metrics nearly unchanged. It also outperforms TaylorSeer in latency and avoids the stronger quality degradation observed with Fast3Dcache. Fig. 13 further shows that Fast-TRELLIS preserves visual quality across both single-view and multi-view examples.

*Table 8.* Transferability evaluation of Fast-TRELLIS on Toys4K.

| Method | CD$_\downarrow$ | $F_1$@0.05$_\uparrow$ | vIoU$_\uparrow$ | Latency (s)$_\downarrow$ | Memory (GB)$_\downarrow$ |
|---|---|---|---|---|---|
| TRELLIS | **0.0635** | **57.19** | 0.295 | 7.68 (1.00×) | 10.38 |
| +TaylorSeer | 0.0638 | 57.01 | 0.299 | 4.65 (1.65×) | 10.40 |
| Fast3Dcache | 0.0658 | 55.69 | 0.248 | 7.91 (0.97×) | 10.52 |
| Fast-TRELLIS | 0.0637 | 57.15 | **0.300** | **3.40 (2.26×)** | **9.97** |

## B.2. GPU Memory Analysis

Fast-SAM3D caches intermediate states in the diffusion stages, but it also shortens the effective sequence length through SLaT token carving and reduces mesh-decoder workload through spectral-aware aggregation. Tab. 9 reports the overall and stage-wise peak memory during inference. Fast-SAM3D reduces the end-to-end peak memory from 19.07GB to 17.89GB, because the original peak is dominated by the mesh stage and our adaptive aggregation directly reduces that workload.

*Table 9.* Overall and stage-wise peak GPU memory during inference.

| Method | Overall Peak (GB)$_\downarrow$ | SS Peak | SLaT Peak | Mesh Peak |
|---|---|---|---|---|
| SAM3D | 19.07 | 13.38 | 13.83 | 19.07 |
| Fast-SAM3D | **17.89** ($-1.18$) | 13.39 ($+0.01$) | **13.43** ($-0.40$) | **17.89** ($-1.18$) |

## C. More Analysis of Modality-Aware Step Caching

In this section, we provide a deeper investigation into the motivation behind our Modality-Aware Step Caching strategy.

### C.1. Visualizing Trajectory Heterogeneity

To empirically validate the *modality heterogeneity* assumption discussed in Sec. 4.1, we visualize the denoising trajectories of different token types in the latent space. Fig. 8 plots the evolution of representative Shape Tokens ($\mathbf{v}^{\text{shape}}$) and Layout Tokens ($\mathbf{v}^{\text{layout}}$) across timesteps.

**Observation & Insight.** As shown in the left column of Fig. 8, **Shape Tokens** exhibit a highly smooth, quasi-linear evolution trajectory. The gradients between adjacent steps change gradually, forming a predictable arc. This high temporal coherence justifies our use of *Linear Extrapolation* for structural tokens, as the local trend at step $t$ is a reliable predictor for step $t - k$. In stark contrast, the right column reveals that **Layout Tokens** undergo volatile, non-monotonic fluctuations. The trajectory is characterized by high-frequency "zig-zag" oscillations. Applying naive linear extrapolation here would amplify these instantaneous fluctuations, leading to severe divergence. This observation strongly supports our design of *Momentum-Anchored Smoothing*, which utilizes the stable anchor from the backbone to dampen these high-frequency jitters while tracking the global pose evolution.

## D. More Analysis of Joint Spatiotemporal Token Carving

In this section, we provide a visual verification of our saliency mechanism's predictive accuracy and analyze the sensitivity of the carving factor $\gamma$.

### D.1. Visualizing Saliency Effectiveness

To validate the reliability of our token carving strategy, we compare our estimated importance score against the 'oracle' ground truth. In Fig. 9, the **'Real Change'** (Left) represents the *ground truth* magnitude of the actual token updates at the next step ($||\mathbf{v}_t - \mathbf{v}_{t-1}||$), indicating where computation is truly needed. The **'Unified Saliency'** (Right) is our predicted importance map ($\mathcal{J}_t$) derived from spatiotemporal history. As observed in the visualization, our Unified Saliency map exhibits a strong correlation with the Ground Truth change map:

- **Accurate Localization:** Our method correctly highlights the regions that subsequently undergo significant updates

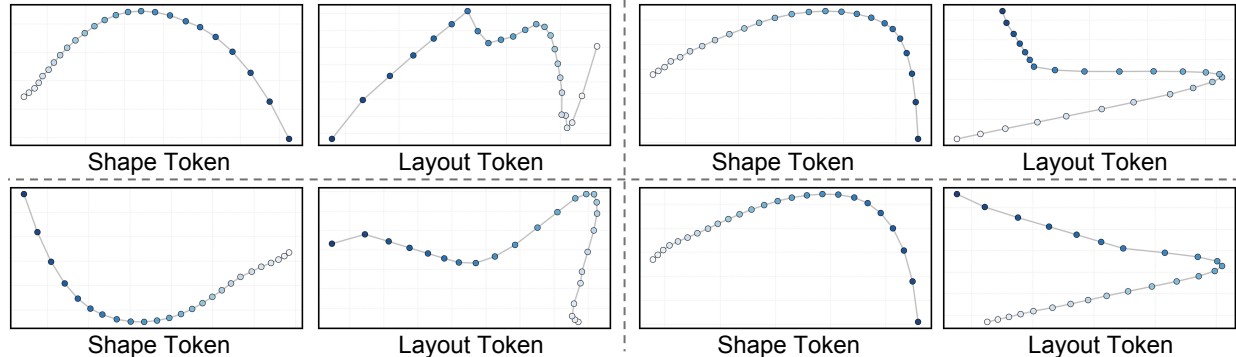

*Figure 8.* **Visualization of latent trajectories.** We visualize the evolution of randomly selected Shape Tokens (Left) and Layout Tokens (Right) during the denoising process. Shape tokens show smooth, predictable evolution, while layout tokens exhibit erratic, high-frequency oscillations, necessitating our Modality-Aware caching strategy.

(e.g., the edges and complex topological structures), effectively capturing the "active set" of the diffusion process.

- **Correctness of Prediction:** The strong visual alignment confirms that our proposed metric, which combines first/second-order temporal dynamics with spatial frequency is a highly accurate proxy for future variation. This ensures that we are not randomly pruning, but selectively preserving tokens that are mathematically destined to change, thereby guaranteeing geometric fidelity even under high sparsity.

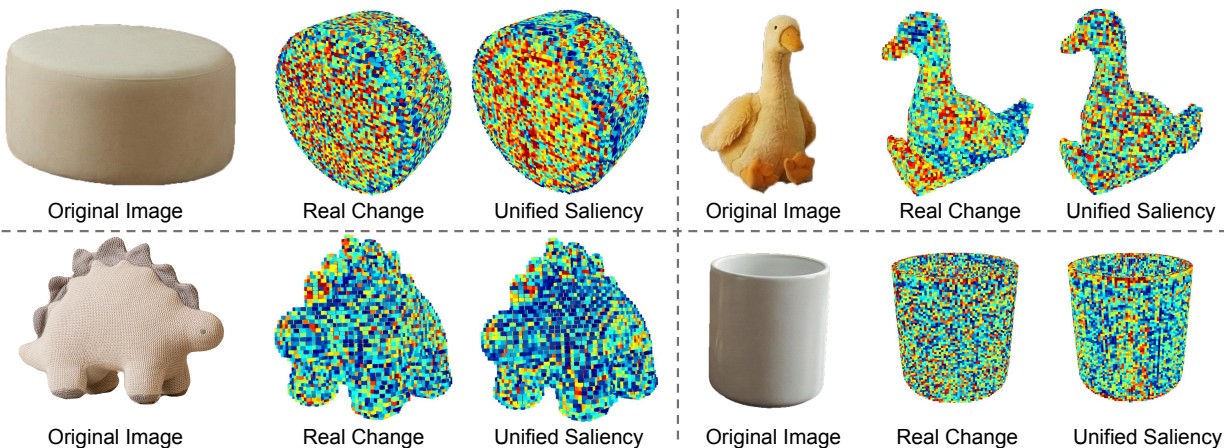

*Figure 9.* **Validation of Saliency Prediction.** We compare the Ground Truth update magnitude (Left) against our Predicted Unified Saliency map (Right). The high consistency between our prediction and the actual future change demonstrates the correctness of our saliency estimation, ensuring that critical computation regions are accurately identified.

*Table 10.* Ablation study on $\gamma$ in Eq. 6.

| $\gamma$ | Uni3D$_\uparrow$ | CD$_\downarrow$ | $F_1$@0.05$_\uparrow$ | 3D-IoU$_\uparrow$ |
|---|---|---|---|---|
| 0.0 | 0.3373 | 0.0215 | 92.061 | 0.375 |
| 0.5 | 0.3474 | 0.0216 | 92.064 | 0.375 |
| 0.7 | 0.3503 | **0.0215** | **92.585** | **0.375** |
| 1.0 | **0.3539** | 0.0217 | 91.759 | 0.375 |

### D.2. Sensitivity to Carving Factor $\gamma$

Tab. 10 assesses the balance between update magnitude and abruptness in our saliency potential (Eq. 6). Setting $\gamma = 0.7$ achieves the best balance. Neglecting abrupt changes ($\gamma = 0.0$) misses critical "turning points" in the diffusion trajectory,

lowering $F_1$. However, over-emphasizing second-order variations ($\gamma = 1.0$) introduces noise sensitivity. The optimal value $\gamma = 0.7$ confirms that while the magnitude of evolution is the primary signal, incorporating derivative changes is essential for capturing transient high-frequency details.

## E. More Analysis of Spectral-Aware Dynamic Token Aggregation

In this section, we provide the empirical motivation for our adaptive aggregation strategy and analyze the contribution of different spectral cues.

### E.1. Visualizing Spectral Heterogeneity

The core premise of our mesh decoding acceleration is that geometric information is non-uniformly distributed across different object instances. To validate this, Fig. 10 visualizes the frequency spectrum distributions of a "Simple Object" (e.g., a smooth cylinder) and a "Complex Object" (e.g., a dragon).

**Spectrum-Information Gap.** As shown in the visualization:

- **Simple Objects (Left):** The spectral energy is highly concentrated in the low-frequency band and decays rapidly. This indicates that the object's geometry is dominated by smooth surfaces with minimal high-frequency detail. Consequently, applying aggressive token aggregation (e.g., a coarse grid) incurs negligible information loss while maximizing speed.

- **Complex Objects (Right):** In contrast, the spectrum exhibits a "long-tail" distribution with significant energy persisting in higher frequencies. This signifies the presence of intricate topological details and sharp edges. A uniform downsampling strategy would severely alias these high-frequency components, leading to the loss of fine geometry.

This distinct spectral heterogeneity strongly justifies our **Instance-Adaptive** design: explicitly measuring the High-Frequency Energy Ratio (HFER) allows us to dynamically allocate computational budget (grid resolution) where it is structurally needed.

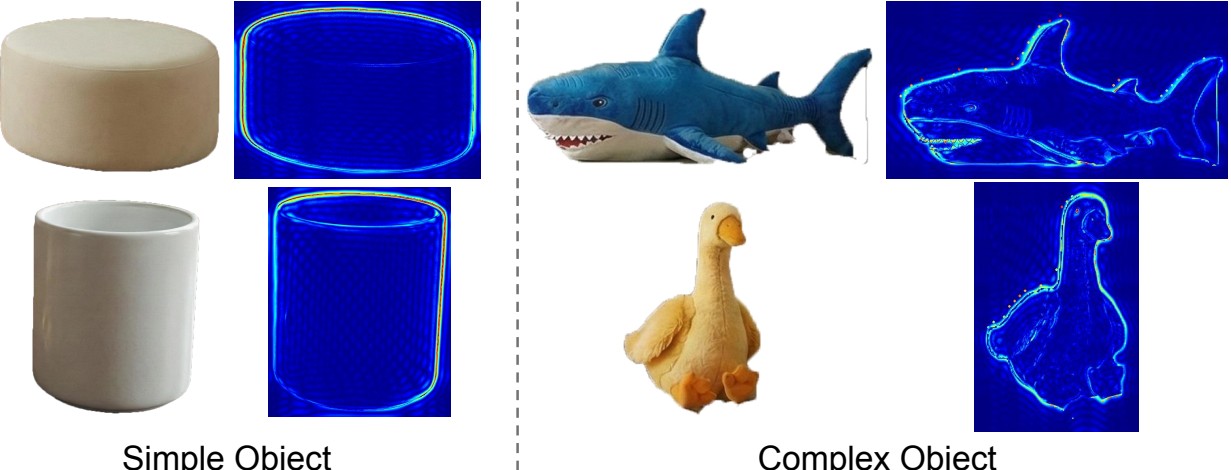

Simple Object     Complex Object

*Figure 10.* **Spectral Analysis of Geometric Complexity.** We compare the Fourier spectrum of a simple object (left) versus a complex object (right). Simple objects show rapid energy decay, supporting aggressive aggregation, whereas complex objects retain significant high-frequency energy, necessitating a finer token grid.

### E.2. Sensitivity to Complexity Weight $w$

Our complexity metric $\mathcal{H}_{\text{joint}}$ fuses cues from the 2D silhouette and the coarse 3D voxel grid, balanced by weight $w$. Tab. 11 investigates the impact of this fusion.

**The Role of 2D vs. 3D Cues.**

*Table 11.* Ablation study on $w$ in Eq. 15.

| $w$ | Uni3D$_\uparrow$ | CD$_\downarrow$ | $F_1$@0.05$_\uparrow$ | 3D-IoU$_\uparrow$ |
|---|---|---|---|---|
| 0.0 | 0.3194 | 0.0213 | 92.557 | 0.375 |
| 0.5 | 0.3163 | 0.0215 | 92.552 | 0.375 |
| 0.9 | **0.3503** | 0.0215 | **92.585** | 0.375 |
| 1.0 | 0.3474 | **0.0211** | 92.103 | 0.375 |

- **Failure of Pure 3D** ($w = 0.0$)**:** Relying solely on the coarse voxel grid results in the lowest perceptual score (Uni3D: 0.3194). This is expected because the input voxel from Stage 1 is inherently 'coarse' and lacks the high-frequency surface details required for fine complexity estimation.

- **Risks of Pure 2D** ($w = 1.0$)**:** While the 2D mask captures sharp boundaries, relying on it entirely ($w = 1.0$) neglects internal volumetric topology, leading to a drop in geometric precision ($F_1$ decreases to 92.103).

- **Optimal Fusion** ($w = 0.9$)**:** The results show a clear preference for 2D dominance ($w = 0.9$), achieving the best balance (Uni3D: **0.3503**, $F_1$: **92.585**). This suggests that the high-resolution 2D input provides the primary signal for detail density, while the 3D voxel acts as a necessary topological regularizer to prevent misjudging voluminous but smooth shapes.

## F. Robustness Analysis

In this section, we include additional robustness studies from the rebuttal. These experiments stress Fast-SAM3D from three complementary perspectives: broader ADT view coverage, degraded mask inputs, and repeated regeneration/evaluation.

### F.1. Broader ADT View Coverage

The main paper evaluates scene layout alignment on 16 ADT views. To test whether the conclusion holds under broader view coverage, we further expand the ADT evaluation to 128 views. As shown in Tab. 12, Fast-SAM3D remains near-lossless compared with SAM3D and is stronger than TaylorSeer overall: it matches TaylorSeer's 3D-IoU, obtains lower ICP-rot error, and preserves the same 2.67× object-level acceleration.

*Table 12.* Ablation study on expanded ADT evaluation with 128 views.

| **Method** | 3D-IoU$_\uparrow$ | ICP-rot$_\downarrow$ | Object Time(s) $_\downarrow$ |
|---|---|---|---|
| SAM-3D | **0.403** | **18.11** | 31.04 |
| TaylorSeer | 0.401 | 18.98 | 22.93 (1.35×) |
| **Fast-SAM3D** | 0.401 | 18.53 | **11.60 (2.67×)** |

### F.2. Robustness to Degraded Mask Inputs

Fast-SAM3D uses the 2D mask in the spectral-aware aggregation module, so we further test a hard setting where 50% of the major object region is masked out on Toys4K. We also sweep the anchoring coefficient $\beta$ around the default setting to check whether low-quality masks and coefficient perturbations cause unstable behavior. Tab. 13 shows that Fast-SAM3D remains stable across $\beta = 0.4/0.5/0.6$, stays close to SAM3D in quality, improves over TaylorSeer, and retains the 11.60s latency.

### F.3. Reproducibility Across Runs

To verify that the reported gains are not incidental, all experiments are based on five rounds of regeneration and evaluation. Tab. 14 reports the mean and standard deviation across runs. Fast-SAM3D exhibits small variance on both object-level geometry and scene-level layout metrics, supporting the reproducibility of the speed-quality trade-off.

*Table 13.* Ablation study under degraded masks on Toys4K, where 50% of the major object region is masked out.

| Method | $\beta$ | CD$_\downarrow$ | $F_1$@0.05$_\uparrow$ | vIoU$_\uparrow$ | Latency(s) $_\downarrow$ |
|---|---|---|---|---|---|
| SAM-3D | – | 0.024 | **90.28** | 0.504 | 31.04 |
| TaylorSeer | – | 0.024 | 89.83 | 0.506 | 22.93 |
| Fast-SAM3D | 0.4 | 0.024 | 89.92 | 0.511 | **11.60** |
| **Fast-SAM3D** | **0.5** | 0.024 | 89.93 | **0.512** | **11.60** |
| Fast-SAM3D | 0.6 | 0.024 | 89.91 | 0.510 | **11.60** |

*Table 14.* Ablation study of statistical variation across five regeneration/evaluation runs.

| Method | Uni3D$_\uparrow$ | CD$_\downarrow$ | $F_1$@0.05$_\uparrow$ | vIoU$_\uparrow$ | 3D-IoU$_\uparrow$ | ICP-rot$_\downarrow$ |
|---|---|---|---|---|---|---|
| SAM-3D | 0.369±0.002 | 0.022±0.001 | 92.34±0.025 | 0.543±0.001 | 0.403±0.01 | 19.32±0.10 |
| **Fast-SAM3D** | 0.350±0.003 | 0.022±0.001 | **92.59**±0.021 | **0.552**±0.002 | 0.375±0.01 | **17.71**±0.12 |

# G. Reproducibility Statement

To enhance reproducibility, we have attached our necessary code and the generated raw mesh files in the supplementary material.

# H. More visual comparison

Here, we provide more visual comparisons to demonstrate the effectiveness of our proposed **Fast-SAM3D**. The results are shown in Fig. 11 and Fig. 12.

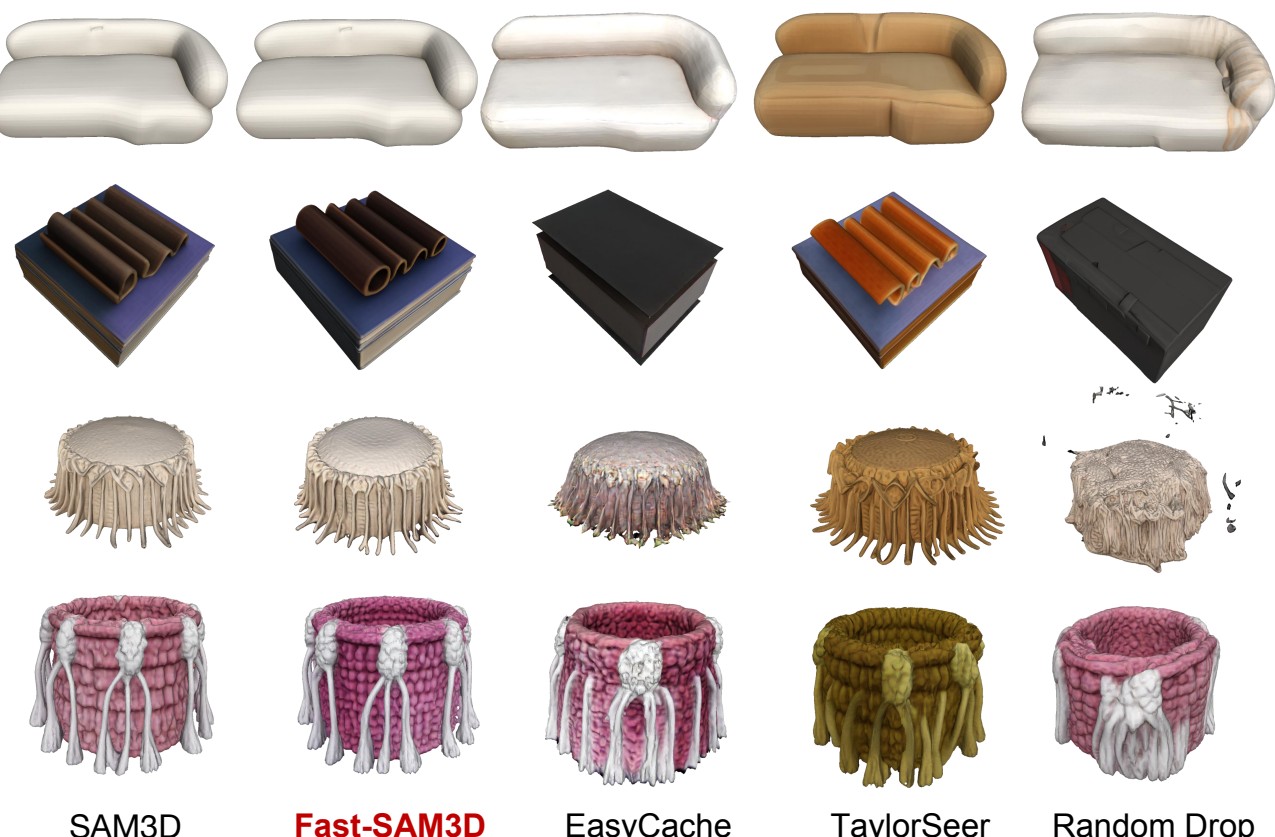

| SAM3D | **Fast-SAM3D** | EasyCache | TaylorSeer | Random Drop |

*Figure 11.* More visual comparison between **Fast-SAM3D** and existing methods.

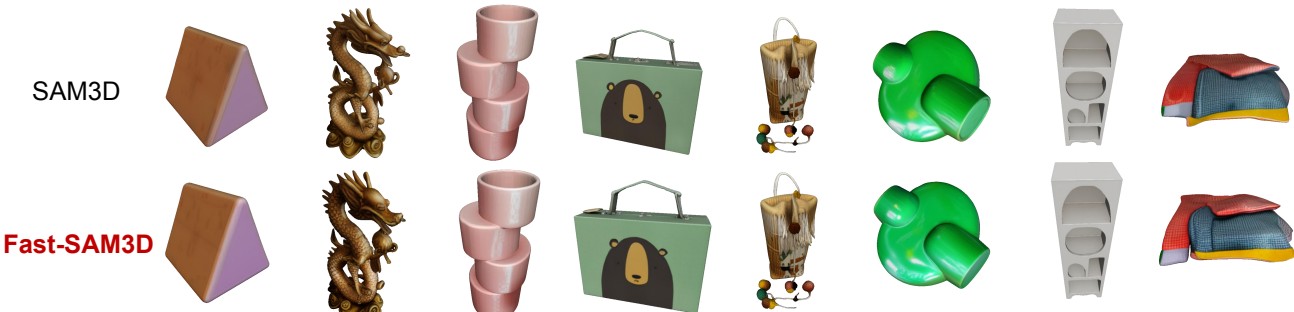

*Figure 12.* More visual comparison between **Fast-SAM3D** and original SAM3D (Chen et al., 2025).

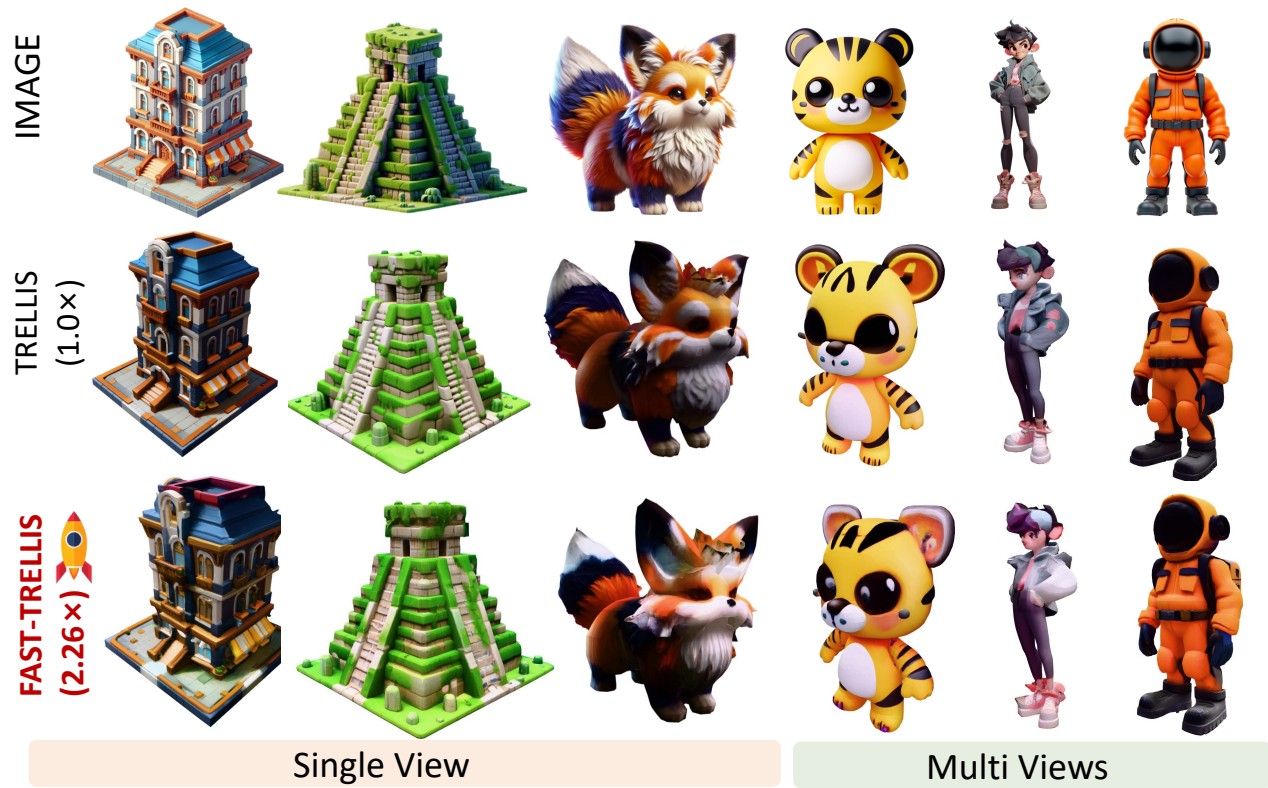

*Figure 13.* **Qualitative comparison on TRELLIS.** Fast-TRELLIS transfers the heterogeneity-aware acceleration principle of Fast-SAM3D to TRELLIS and achieves a 2.26× speedup while preserving visual quality across single-view and multi-view reconstruction examples.

