# OpenReview forum: "Fast-SAM3D: 3Dfy Anything in Images but Faster"
_ICML.cc/2026/Conference — ICML 2026 regular_

### Official Review · Reviewer_6RiH · 2026-03-10

**Soundness:** 3
**Presentation:** 3
**Significance:** 3
**Originality:** 3
**Overall Recommendation:** 4
**Confidence:** 3

**Summary:**

This paper conducts a systematic performance characteristic analysis on the problem of excessively high inference cost of the existing 3D reconstruction model SAM3D. It is found that the common acceleration strategies are prone to fail when dealing with this task. The authors point out that this is mainly due to the multi-level heterogeneity in the diffusion pipeline: including kinematic differences in shape and layout, the inherent sparsity in the texture refinement process, and the significant differences in the frequency domain of different geometric structures. Experimental results show that this method achieves an object-level acceleration ratio of up to 2.67 times without sacrificing the reconstruction accuracy.

**Compliance With Llm Reviewing Policy:**

Affirmed.

**Final Justification:**

Thank you for your detailed rebuttal and additional experiments. This has addressed my concerns. This manuscript is worthy of acceptance.

**Key Questions For Authors:**

1. Due to the introduction of Modality-Aware Step Caching and Joint Spatiotemporal Token Carving, the system needs to cache the features or tangent offset $\Delta_{i}$ of the anchor point steps. How much has the peak memory usage of Fast-SAM3D during actual inference increased compared to the baseline SAM3D? Will this become a bottleneck on resource-constrained devices?
2. Will the spectral sensing aggregation strategy cause excessive resolution compression on objects with numerous fine and high-frequency details?
3. If there is extreme perspective distortion in the image and the momentum anchoring coefficient is improperly set (such as affected by cumulative errors) resulting in early global coordinate drift, does the model have a mechanism in subsequent layers to correct this deviation?
4. In the spectral sensing Token aggregation module of the model, it highly relies on the high-frequency energy ratio $\mathcal{H}(M_{2D})$ of the input 2D Mask to evaluate geometric complexity and determine the downsampling rate $\mathcal{S}$. The standard input of SAM3D includes images and Masks. In actual open-world applications, user interaction or the automatic segmentation model generating Masks often have jagged edges or noise interference. Is this artificial high-frequency noise caused by poor Mask quality likely to cause the system to misjudge the complexity of the object, thereby leading to acceleration failure?

**Limitations:**

The main limitation of this study lies in its highly customized approach to a specific two-stage architecture in the 3D diffusion pipeline. Although the three proposed modules performed exceptionally well under the SAM3D framework, their design depth was tightly bound to the sparse structure (SS) and sparse latent variable (SLaT) cascading generation paradigm of SAM3D. This makes it challenging to directly transfer them to a single-stage 3D generation network.

**Strengths And Weaknesses:**

Strengths:

1. The article not only presents the solution, but also conducts a comprehensive modularized analysis of the reasoning dynamics of SAM3D.
2. As an end-to-end training-free framework, Fast-SAM3D seamlessly integrates into the existing inference process. This significantly lowers the implementation threshold.
3. A large number of experiments have proved that this method significantly outperforms baseline models such as TaylorSeer and EasyCache.


Weaknesses:

1. This framework introduces a number of key scalar parameters, such as the cache stride $k$, the momentum coefficient $\beta$, the adaptive switching threshold $\mathcal{E}$, and the spectral thresholds for grid merging $\{\tau_{low}, \tau_{high}\}$, etc.
2. The acceleration of the article heavily relies on the caching of features and tangent updates (Cached Update, Tangent Reuse). Frequent caching of high-dimensional Tokens and historical offsets may lead to an increase in peak memory usage.
3. The spectral-aware aggregation module relies on high-quality 2D Mask inputs to calculate the high-frequency energy ratio $\mathcal{H}(M_{2D})$. In unconstrained scenarios with severe occlusion or shadows, if the segmentation model provides low-quality Masks, it may lead to incorrect judgments in the aggregation strategy.

---

> ### Author Rebuttal · Authors · 2026-03-31
>
> Thank you for the careful review and your helpful suggestions. We respond to your questions as follows.
>
> >**W1: Introducing key scalar parameters.**
>
> Thank you for pointing this out. But we want to highlight that the method is **not tuned around a razor-thin optimum**. In the paper, we already provide ablations for the main scalar parameters, including the cache stride $k$ in Tab. 6, momentum coefficient $\beta$ in Tab. 4, switching threshold $\varepsilon$ in Tab. 8, spectral thresholds $\tau_{low}, \tau_{high}$ in Tab. 5, and more other parameters. Some results are below:
>
> - **$\beta$** in Table 4 is stable across `0.3-0.9`, keeping **F1 within 92.57-92.59**;
> - **k=2/3** in Table 6 both retain **>92.58 F1**;
> - **$\varepsilon$** in Table 8 across `1.0-2.0`  keeps **vIOU > 0.549**;
> - **E=1.0-2.0** in Table 8 changes runtime only mildly, with the best balance at **E=1.5**.
>
> These results show that the method is reasonably stable within a meaningful range, and that the final configuration is not tuned around a narrow optimum.
>
> >**W2 and Q1: Peak memory usage.**
>
> We further conducted memory usage ablation study, which shows that we can **even reduce peak memory**. Details please see our rebuttal to **Reviewer 4dhL `W2 and Q1: GPU memory analysis`**.
>
> >**W3, Q3, and Q4: Robustness of 2D masks and under extreme perspective distortion in the image and the momentum anchoring coefficient.**
>
> Thank you for this important comment. We agree that, in extreme open-world cases, low-quality 2D masks may introduce extra noise. However, our spectral-aware aggregation is **not** driven by the 2D mask alone.
>
> First, complexity is estimated jointly from the 2D silhouette and the coarse 3D voxel cue:
>
> $$
> \mathcal{H}\_{\mathrm{joint}} = w\cdot \mathcal{H}(\mathbf{M}\_{\mathrm{2D}}) + (1-w)\cdot \mathcal{H}(\mathbf{V}\_{\mathrm{3D}})
> $$
>
> Table 10 supports this:
>
> $\omega$|Uni3D↑|CD↓|F1\@0.05↑|3D-IoU↑
> -|-|-|-|-
> 0.0|0.3194|0.0213|92.557|0.375
> 0.5|0.3163|0.0215|92.552|0.375
> 0.9|**0.3503**|0.0215|**92.585**|**0.375**
> 1.0|0.3474|**0.0211**|92.103|0.375
>
> Both pure 2D and 3D yields suboptimal results and the joint setting (`w=0.9`) gives the best balance. Thus, 2D contributes boundary detail, while 3D regularizes spurious high-frequency noise.
>
> Second, the induced policy change is bounded. Eq. 16 maps $\mathcal{H}_{\mathrm{joint}}$ to only three aggregation levels $S \in \{1.25, 1.5, 2.0\}$, so moderate mask perturbations can only induce limited changes in decoding rate rather than catastrophic over-compression.
>
> Third, **we directly test this extreme mode on Toys4K by masking out 50% of the major object**:
>
> Method|Anchoring Coefficient（β）|CD↓|F1\@0.05↑|vIOU↑|Latency (s)↓
> -|-|-|-|-|-
> SAM3D|-|0.024|90.28|0.504|31.04
> TaylorSeer|-|0.024|89.83|0.506|22.93
> Fast-SAM3D|0.4|0.024|89.92|0.511|11.60
> **Fast-SAM3D**|0.5|**0.024**|**89.93**|**0.512**|**11.60**
> Fast-SAM3D|0.6|0.024|89.91|0.510|11.60
>
> Across `β=0.4/0.5/0.6`, Fast-SAM3D remains very stable and clearly improves over TaylorSeer at much lower latency. At the default `β=0.5`, Fast-SAM3D stays close to SAM3D in quality while preserving a `2.67x` speedup.
>
> This new hard-case ablation directly supports robustness to imperfect coefficient settings under complex conditions. Even with degraded masks and `β` swept over `0.4/0.5/0.6`, the method remains stable, which suggests that strong early drift is hard to trigger in practice. This is also consistent with the paper's original ablations: **our design is not sensitive to β**, indicating that the default setting already lies in a stable regime.
>
> Overall, the new experiment suggests that the current design is robust and minimizes acceleration-induced degradation; if early error does appear, the later coarse-to-fine refinement can further reduce its effect in practice.
>
>
> >**Q2: Resolution compression on objects with numerous details.**
>
> In principle, this is exactly what our spectral-aware aggregation is designed to avoid. Eq. 16 assigns a **smaller aggregation factor to high-complexity cases** and therefore retains more decoding resolution. The purpose of the spectral-aware policy is to allocate computation adaptively, so that simple objects are compressed more aggressively while detail-rich objects are preserved more carefully. Table 5 shows that the default thresholds `{0.5, 0.7}` retain strong geometry (**F1 92.585, vIoU 0.5521**) while reducing latency.
>
>
> >**Limitation: Designed for two-stage backbones.**
>
> We agree that the current implementation is most natural for multi-stage backbones such as SAM3D. Importantly, as shown in rebuttal to `Reviewer BU6t, W1`, we also transfer Fast-SAM3D to TRELLIS and obtain 2.26x speedup with comparable quality indicating that the core acceleration principle is not tied to a single backbone. For single-stage generators, whether single-forward or diffusion-based, the adaptation needs to follow their own computation patterns, as noted in the rebuttal to `Reviewer vS6Z, W1 and Q1`; we leave this as future work.

---

> > ### Author Rebuttal · Reviewer_6RiH · 2026-04-03
> >
> > Thank you for your detailed rebuttal and additional experiments. This has addressed my concerns. This manuscript is worthy of acceptance.

---

> > > ### Author Response · Authors · 2026-04-06
> > >
> > > Dear reviewer,
> > >
> > > We sincerely appreciate your time and thoughtful feedback throughout the review process. We are delighted to hear that your concerns have been addressed and that you **consider the manuscript worthy of acceptance**. We are truly grateful for your strong support and valuable evaluation of our work.
> > >
> > > Best wishes,
> > >
> > > All authors

---

### Official Review · Reviewer_4dhL · 2026-03-11

**Soundness:** 3
**Presentation:** 2
**Significance:** 3
**Originality:** 3
**Overall Recommendation:** 5
**Confidence:** 3

**Summary:**

This paper presents Fast-SAM3D, an end-to-end, training-free acceleration framework designed for single-view 3D reconstruction. The authors address the high inference latency of state-of-the-art models by introducing a multi-faceted efficiency strategy that targets diffusion steps, token redundancy, and decoding resolution. The proposed method achieves a significant speedup (up to 2.67x) while maintaining reconstruction quality.

**Compliance With Llm Reviewing Policy:**

Affirmed.

**Final Justification:**

Since the author addressed my question, I stick to my positive score. I look forward to the improved readability of this paper in the updated version.

**Key Questions For Authors:**

My main concern is whether this paper exhibits a "space-for-time" tradeoff, meaning it requires more GPU memory or caching during inference. Excessive GPU memory requirements could potentially impact the method's usability.

**Limitations:**

The modules described in the Methodology section do not always align with the terminology used in the ablation study.

**Strengths And Weaknesses:**

### Strengths
1. Novelty in Token Carving. Using Fast Fourier Transform (FFT) to quantify spatial frequency complexity and combining it with temporal update dynamics provides a principled way to identify "active" regions. This spectral-domain insight effectively captures high-frequency details (complex objects) that require dense computation, while aggressively pruning redundant tokens in smooth areas, which is a major contribution.
2. Solid Engineering Execution. Modality-Aware Step Caching demonstrates impressive engineering maturity.
3. Training-free Efficiency. The framework achieves substantial throughput gains without requiring expensive retraining or fine-tuning, pushing the Pareto frontier of the speed-accuracy trade-off in 3D reconstruction tasks.

### Weakness
1. Readability. The modules described in the Methodology section do not always align with the terminology used in the ablation study. For instance, a module referred to by its full functional name in the text is often abbreviated differently in the tables. It is suggested to replace the generic module abbreviations in the Ablation Study tables with the specific method/algorithm abbreviations
2. GPU Memory Analysis. While the paper focuses heavily on time-based speedup, it misses a critical discussion on memory efficiency. I suggest the authors also include a detailed analysis of GPU memory consumption in the ablation study.

---

> ### Author Rebuttal · Authors · 2026-03-31
>
> We are grateful for your positive assessment of our contributions and for your helpful suggestions. We respond to your questions as follows.
>
> >**W1 and Limitation: Readability about module abbreviations.**
>
> Thank you for this constructive comment. We agree that the terminology used in the Methodology section and the ablation study tables is not fully aligned. In the revised paper, we will replace the generic module abbreviations with the specific method/algorithm abbreviations to improve consistency and readability.
>
> >**W2 and Q1: GPU memory analysis.**
>
> Thanks for your advice. We agree that our method introduces additional cached states in the SS and SLat stages, which brings a small amount of extra memory in these diffusion-related modules. However, this is only one side of the overall memory picture.
>
> A key point is that Fast-SAM3D also introduces **token carving in SLat** and **resolution downsampling in the mesh stage**, both of which shorten the effective sequence length by later modules. These mechanisms reduce the amount of activation memory needed during inference and even **make the overall pipeline more memory-friendly**.
>
> We additionally measure the GPU memory usage during inference, including overall and stage-wise peak memory:
>
> Method|Overall Peak (GB)↓|SS Peak|SLat Peak|Mesh Peak
> -|-|-|-|-
> SAM3D|19.07|**13.38**|13.83|19.07
> **Fast-SAM3D**|**17.89 (-1.18)**|13.39 (+0.01)|**13.43 (-0.40)**|**17.89 (-1.18)**
>
> As shown above, **Fast-SAM3D does not increase the peak inference memory relative to SAM3D; instead, it even reduces the overall peak memory from 19.07 GB to 17.89 GB**. Importantly, the stage-wise breakdown reveals that the peak memory of the original SAM3D pipeline is **dominated by the mesh stage**, rather than by the diffusion-stage feature states. Since Fast-SAM3D reduces the effective workload in the mesh stage through spectral-aware downsampling, it correspondingly reduces the dominant source of peak memory.
>
> Overall, our Fast-SAM3D improves runtime efficiency without introducing peak memory; instead, because the original peak memory is mesh-dominated, Fast-SAM3D actually reduces the end-to-end peak GPU memory.

---

> > ### Author Rebuttal · Reviewer_4dhL · 2026-04-03
> >
> > Since the author addressed my question, I stick to my positive score. I look forward to the improved readability of this paper in the updated version.

---

> > > ### Author Response · Authors · 2026-04-06
> > >
> > > Dear reviewer,
> > >
> > > We sincerely appreciate your time and constructive feedback throughout the review process. We are delighted to hear that **your concern has been addressed** and that you continue to support the paper with a positive score. We will improve the readability in the updated version. We are very grateful for your encouragement and support.
> > >
> > > Best wishes,
> > >
> > > All authors

---

### Official Review · Reviewer_vS6Z · 2026-03-12

**Soundness:** 3
**Presentation:** 3
**Significance:** 2
**Originality:** 3
**Overall Recommendation:** 4
**Confidence:** 3

**Summary:**

This paper presents Fast-SAM3D, a training-free acceleration framework for SAM3D that targets three identified bottlenecks in the pipeline: sparse structure generation, SLaT generation, and mesh decoding. The method combines modality-aware step caching, token carving with adaptive caching, and spectral-aware token aggregation, and reports substantial speedups with limited quality loss on the evaluated benchmarks.

**Compliance With Llm Reviewing Policy:**

Affirmed.

**Final Justification:**

Thank you for the detailed rebuttal. My concerns have been addressed, and I am happy to keep my positive rating.

**Key Questions For Authors:**

1. How much of the proposed approach would transfer beyond SAM3D to other single-view 3D generation or diffusion-based reconstruction pipelines, and which components are fundamentally SAM3D-specific? More insights on this would be appreciated.

2. Can the method be simplified into a smaller number of more unified design principles, rather than relying on several stage-specific heuristics and mechanisms?

3. How robust are the reported gains under broader evaluation, such as more ADT scenes/views, additional datasets, or settings with stronger geometric ground truth?

4. Would the conclusions remain the same under stronger validation of quality preservation, including statistical variation across runs and more direct geometry-sensitive metrics?

**Limitations:**

yes

**Strengths And Weaknesses:**

Strengths

* Concrete analysis and practical motivation. The paper clearly identifies where latency arises in SAM3D and connects each bottleneck to a targeted intervention, which makes the work easy to follow and practically relevant for this system.
* Meaningful empirical speedups on the chosen benchmark. The reported results show large reductions in runtime relative to the SAM3D baseline, while keeping the main quality metrics close and in some cases slightly improved in Table 2. That makes the paper useful from an engineering perspective.
* Ablations supporting the component design. Ablations for module combinations, momentum factors, and aggregation thresholds provide evidence to support the effectiveness of the proposed method.

Weaknesses

* Limited breadth beyond one specific model. The paper is framed around accelerating SAM3D, and most of the motivation, diagnosis, and solution design remain tightly coupled to that architecture and its internal stages, instead of addressing a broader question in 3D generation or efficient diffusion more generally. As a result, the contribution reads more like a system-specific optimization study than a broadly transferable scientific advance.
* The solution is fragmented into multiple ad hoc components. While the analysis is concrete, it ultimately leads to several small mechanisms including different caching rules, token carving, adaptive schedules, and aggregation heuristics, which are combined to fix different local inefficiencies in SAM3D. This makes the overall method feel somewhat patchwork, and it is harder to extract a simple, generalizable principle from the paper.
* Evaluation scope. The experiments show a convincing gain on the reported setup, but the benchmark coverage is relatively limited, with only 16 ADT views and an ISO3D evaluation based on a proxy perceptual similarity rather than direct 3D geometric supervision.

I rated the paper a fair in Significance because, although the idea is technically reasonable, the contribution remains narrowly scoped to a specific model.

---

> ### Author Rebuttal · Authors · 2026-03-31
>
> We are deeply grateful for your support of our work, and we provide detailed responses to your comments as follows:
>
> >**W1 and Q1: More evaluation on other backbones and regarding transferability.**
>
> Thank you for raising this important point. For more evaluation on other backbones, please see our rebuttal to **Reviewer BU6t** `W1: More evaluation on other backbones`. **The TRELLIS result there provides direct empirical evidence beyond SAM3D**, here we clarify why such transfer is feasible and what exactly transfers.
>
> Our core motivation is to accelerate a broader class of **highly generalizable 3D reconstruction foundation models**, for which SAM3D serves as a representative example. These models typically rely on large generative priors and iterative generation to recover plausible 3D content, which leads to substantial computation overhead. Our work is motivated by this broader bottleneck rather than by the specifics of a single architecture. The most precise boundary is therefore:
> * **Transferable**: redundancy diagnosis + operator family;
> * **Backbone-specific**: exact insertion points, schedules, and thresholds.
>
> In this sense, our method can be easily transferred to **related backbones with similar iterative denoising and decoding bottlenecks**.
>
> The module-level mapping is summarized below:
>
> Module|Computational bottleneck|Acceleration criterion|Transferable setting
> -|-|-|-
> **Modality-Aware Step Caching**|Iterative denoising spends redundant compute across steps|Reduce temporal redundancy across diffusion steps|To other iterative diffusion pipelines
> **Joint Spatiotemporal Token Carving**|Latent refinement spends compute on many low-change regions through iterative updates|Reduce spatial/structural redundancy during latent refinement by focusing compute on active regions|To refinement stages in diffusion-based reconstruction models where updates are spatially sparse
> **Spectral-Aware Token Aggregation**|Structured decoding over many tokens/voxels wastes resolution on geometrically simple instances|Reduce decoding-resolution redundancy by adapting reduction strength to instance complexity.|To structured decoders with varying length tokens/voxels
>
>
> >**W2 and Q2: Generalizable principle of the proposed pipeline.**
>
> Thank you for this important comment. For the broader transferability framing and claim boundary, please see our response to **W1 and Q1 above**. Here, we focus on the unifying principle behind the method design. As summarized in the table above, the three modules correspond to three different computational bottlenecks and three matching acceleration rules.
>
> And our method can be summarized as **allocating computation along the dominant redundancy axis of each stage while preserving the information most critical to quality**. Under this view, our contribution is best understood as **one unified principle with three stage-aware instantiations**, rather than several local fixes.
>
>
> >**W3, and Q3: Broader benchmark and more direct geometry-sensitive metrics.**
>
> To address this concern, we strengthen the evidence along two axes: broader ADT coverage and stronger geometry-grounded evaluation.
>
> - **Broader ADT coverage.** We expand ADT from **16** to **128** views. Under this broader setting, Fast-SAM3D remains near-lossless while still remaining stronger than **TaylorSeer** overall with comparable **3D-IoU**, better **ICP-rot**, and a **2.67x** speedup.
>
> Method|3D-IoU↑|ICP-rot↓|Object Time (s)↓
> -|-|-|-
> SAM-3D|0.403|18.11|31.04
> TaylorSeer|0.401|18.98|22.93 (1.35×)
> **Fast-SAM3D**|**0.401**|**18.53**|**11.60 (2.67×)**
>
> - **Stronger geometry-grounded metrics.** Beyond the proxy perceptual metric **Uni3D** on ISO3D, the paper already reports direct geometry metrics on **Toys4K**: **Chamfer Distance (CD)**, **F1\@0.05**, and **volumetric IoU (vIoU)**. As defined in **Appendix A.2**, these metrics are computed against **ground-truth 3D**, making them a more direct assessment of reconstruction fidelity:
>
> Method|CD↓|F1\@0.05↑|vIoU↑
> -|-|-|-
> SAM-3D|0.022|92.34|0.543
> **Fast-SAM3D**|**0.022**|**92.59**|**0.552**
>
> These results show that Fast-SAM3D's gains remain stable under broader ADT evaluation and are supported by direct ground-truth geometry metrics, rather than only a relative perceptual proxy.
>
> >**Q4: Statistical variation across runs.**
>
> Thank you for this important question. **All our experiment results in the paper are already based on multiple runs** (5 times of regeneration and evaluation) averaged to avoid incidental results. The statistical variation across runs is further reported below, showing that Fast-SAM3D exhibits stable performance with relatively small variances:
>
> Method|Uni3D↑|CD↓|F1\@0.05↑|vIoU↑|3D-IoU↑|ICP-rot↓|
> -|-|-|-|-|-|-|
> SAM-3D|0.369($\pm$ 0.002)|0.022($\pm$ 0.001)|92.34($\pm$ 0.025)|0.543($\pm$ 0.001)|0.403($\pm$ 0.01)|19.32($\pm$ 0.10)|
> **Fast-SAM3D**|0.350($\pm$ 0.003)|0.022($\pm$ 0.001)|92.59($\pm$ 0.021)| 0.552($\pm$ 0.002)|0.375($\pm$ 0.01)|17.71($\pm$ 0.12)|

---

> > ### Author Rebuttal · Reviewer_vS6Z · 2026-04-03
> >
> > Thank you for the detailed rebuttal. My concerns have been addressed, and I am happy to keep my positive rating.

---

> > > ### Author Response · Authors · 2026-04-06
> > >
> > > Dear reviewer,
> > >
> > > We sincerely appreciate your time and thoughtful feedback throughout the review process. We are delighted to hear that your **concerns have been addressed** and that you are happy to keep your positive rating. We are truly grateful for your support and valuable evaluation of our work.
> > >
> > > Best wishes,
> > >
> > > All authors

---

### Official Review · Reviewer_BU6t · 2026-03-13

**Soundness:** 3
**Presentation:** 2
**Significance:** 2
**Originality:** 3
**Overall Recommendation:** 4
**Confidence:** 4

**Summary:**

Fast-SAM3D is a training-free acceleration framework designed to address the high inference latency of SAM3D, a state-of-the-art single-view multi-object reconstruction model. By conducting a systematic investigation into the inference dynamics, the authors identify multi-level heterogeneity across the generation pipeline: kinematic differences between shape and layout evolution, spatial sparsity in texture refinement, and spectral variance across different geometric complexities. To exploit these redundancies, the framework introduces three plug-and-play modules: Modality-Aware Step Caching to decouple structural and layout updates, Joint Spatiotemporal Token Carving to focus computation on high-entropy regions, and Spectral-Aware Token Aggregation to adaptively compress simple geometries. Experimental results demonstrate that Fast-SAM3D achieves up to a 2.67x.

**Compliance With Llm Reviewing Policy:**

Affirmed.

**Key Questions For Authors:**

Please see weakness part for details.

**Limitations:**

Yes

**Strengths And Weaknesses:**

**Strengths:**

1.The motivation for each acceleration module is well-grounded in empirical observations. The paper provides excellent visualizations for analysis, which make the underlying logic intuitive and convincing.
2.Achieving a significant speedup (2.01x for scenes) without any category-specific training or distillation makes this a highly practical solution for real-world deployment.
3.The saliency-aware mechanisms act as a "denoising filter," allowing the model to actually improve geometric metrics like F-Score in some cases by pruning low-confidence noisy tokens.

**Weaknesses:**

1. The impact of the work could be significantly broader. The proposed spatiotemporal token carving and spectral-aware aggregation appear to be general strategies applicable to various voxel-based 3D generators (e.g., TRELLIS). Validating the effectiveness of these modules on other frameworks would provide much stronger evidence of the method’s universality.
2. Figure 7 Caption Error.
3. The content intended to be expressed in the "Object" row in the teaser is unclear
4. The primary bottleneck of the original SAM3D model is often its positional accuracy rather than its speed. Since SAM3D was not originally optimized for efficiency (e.g., via distillation), accelerating a model that still faces fundamental geometric challenges may be seen as a secondary priority compared to improving its basic reconstruction quality.

---

> ### Author Rebuttal · Authors · 2026-03-31
>
> We sincerely thank you for your positive assessment of our work and your constructive suggestions. We address your comments in detail below.
>
> >**W1: More evaluation on other backbones.**
>
> Thank you for this valuable suggestion.
>
> Our method is motivated by inference-time heterogeneity in structured 3D generation, rather than by SAM3D-specific training objectives. Therefore, the key components are in principle portable to other voxel-/latent-based pipelines. To directly address this concern, we additionally migrated the **full Fast-SAM3D framework** to **TRELLIS** [1] and evaluated it on **Toys4K**. This transfer covers the complete inference-time design of Fast-SAM3D, including **Modality-Aware Step Caching**, **Joint Spatiotemporal Token Carving**, and **Spectral-Aware Token Aggregation**.
>
> Method|CD↓|F1\@0.05↑|vIoU↑|Latency (s)↓|GPU Memory (GB)↓
> -|-|-|-|-|-
> TRELLIS|0.0635|57.19|0.295|7.68 (1.00×)|10.38
> +Tayorseer|0.0638|57.01|0.299|4.65 (1.65×)|10.40
> +Fast3Dcache|0.0658|55.69|0.248|7.91 (0.97×)|10.52
> **+Ours**|**0.0637**|**57.15**|**0.300**|**3.40 (2.26×)**|**9.97**|
>
> On TRELLIS, our method reduces inference time from **7.68s to 3.40s (2.26x)** while keeping the main geometry metrics essentially unchanged. It is also faster than **TaylorSeer** on this setup (**4.65s**) and avoids the stronger quality/runtime trade-off seen in **Fast3DCache** (**CD 0.0658, F1 55.69, vIoU 0.248, 7.91s**).
>
> This supports that our framework is not tied to SAM3D only, but supports broader applicability to **related structured, iterative voxel-/latent-based 3D generation backbones**.
>
> [1]. Structured 3D Latents for Scalable and Versatile 3D Generation. (CVPR 2025)
>
> > **W2 and W3: Caption error and teaser figure unclear.**
>
> Thank you for your valueable advise. this out. In the revision, we will make the presentation fixes explicit:
>
> - correct the **Figure 7** caption;
> - clarify the **"Object"** row in the teaser and explain its intent more clearly in the caption;
>
> >**W4: Original SAM3D accuracy.**
>
> We agree that reconstruction quality is fundamental. Our point is **not** that speed matters more than geometry, but that latency is an **independent deployment bottleneck** and should be improved **without sacrificing the main fidelity metrics**.
>
> Table 2 shows exactly this trade-off: scene/object time drops from **462.3s / 31.04s** to **229.7s / 11.60s**, while **CD stays unchanged**, and even **slightly improves F1, vIoU, and ICP-rot**. We therefore view Fast-SAM3D as a **quality-preserving acceleration layer on top of SAM3D**, not as a substitute for future work on improving the backbone's geometric accuracy.
>
> Method|CD↓|F1\@0.05↑|vIoU↑|3d-IoU↑|ICP-rot↓|Scene Time (s)↓|Object Time (s)↓|
> -|-|-|-|-|-|-|-
> SAM-3D|0.022|92.34|0.543|0.403|19.32|462.3|31.04
> **Fast-SAM3D**|**0.022**|**92.59**|**0.552**|0.375|**17.71**|**229.7 (2.01×)**|**11.60 (2.67×)**
>
> Since our method is training-free and applied purely at inference time, it is **orthogonal to future advances** in model quality and can be combined with stronger reconstruction backbones as they become available.
>
> We will revise the paper to make this positioning clearer.

---

> > ### Author Rebuttal · Reviewer_BU6t · 2026-04-05
> >
> > I thank the authors for their response. My concerns have been addressed.

---

> > > ### Author Response · Authors · 2026-04-06
> > >
> > > Dear reviewer,
> > >
> > > We sincerely appreciate your time and constructive feedback throughout the review process. We are delighted to hear that **your concerns have been addressed**. We are grateful for your careful evaluation and valuable support of our work.
> > >
> > > Best wishes,
> > >
> > > All authors

---

### Decision · Program_Chairs · 2026-04-30

**Decision:**

Accept (regular)

**Comment:**

All reviewers are consistently positive. After reading the paper, rebuttal and all comments, the AC also agrees with reviewers.